# Function and firing of the *Streptomyces coelicolor* contractile injection system requires the membrane protein CisA

**Bastien Casu[1], Joseph W Sallmen[2], Peter E Haas[1], Govind Chandra[2], Pavel Afanasyev[3], Jingwei Xu[1], Martin Pilhofer[1]\*, Susan Schlimpert[2,4]\***

[1]Department of Biology, Institute of Molecular Biology & Biophysics, Eidgenössische Technische Hochschule Zürich, Zurich, Switzerland; [2]John Innes Centre, Department of Molecular Microbiology, Norwich Research Park, Norwich, United Kingdom; [3]Cryo-EM Knowledge Hub, Eidgenössische Technische Hochschule Zürich, Zurich, Switzerland; [4]Centre for Microbial Interactions, Norwich Research Park, Norwich, United Kingdom

**\*For correspondence:**
pilhofer@mol.biol.ethz.ch (MP);
susan.schlimpert@jic.ac.uk (SS)

**Competing interest:** The authors declare that no competing interests exist.

## eLife Assessment

This **valuable** study provides insights into the structure and function of bacterial contractile injection systems that are present in the cytoplasm of many Streptomyces strains. A **convincing** high-resolution model of the structure of extended forms of the cytoplasmic contractile injection system assembly from Streptomyces coelicolor is presented, with some investigation of the membrane protein CisA in attachment of the extended assembly to the inner face of the cytoplasmic membrane and the firing of the system. The work expands the current understanding of these diverse bacterial nanomachines.

**Abstract** Bacterial contractile injection systems (CIS) are phage tail-like macromolecular complexes that mediate cell-cell interactions by injecting effector proteins into target cells. CIS from *Streptomyces coelicolor* (CIS[Sc]) are localized in the cytoplasm. Under stress, they induce cell death and impact the *Streptomyces* life cycle. It remains unknown, however, whether CIS[Sc] require accessory proteins to directly interact with the cytoplasmic membrane to function. Here, we characterize the putative membrane adaptor CisA, a conserved factor in CIS gene clusters across *Streptomyces* species. We show by cryo-electron tomography imaging and in vivo assays that CIS[Sc] contraction and function depend on CisA. Using single-particle cryo-electron microscopy, we provide an atomic model of the extended CIS[Sc] apparatus; however, CisA is not part of the complex. Instead, our findings show that CisA is a membrane protein with a cytoplasmic N-terminus predicted to interact with CIS[Sc] components, thereby providing a possible mechanism for mediating CIS[Sc] recruitment to the membrane and subsequent firing. Our work shows that CIS function in multicellular bacteria is distinct from type VI secretion systems and extracellular CIS, and possibly evolved due to the role CIS[Sc] play in regulated cell death.

## Introduction

Bacteria employ different types of contractile injection systems (CIS) to mediate cell-cell interactions or cell death (*Lin, 2024*). CIS are evolutionarily and structurally related to contractile tails of bacteriophages and are comprised of core modules, including a baseplate, a contractile sheath, and an inner tube (*Brackmann et al., 2017*; *Taylor et al., 2018*). CIS firing is triggered by a conformational

change within the baseplate (*Taylor et al., 2016*; *Böck et al., 2017*) and results in the contraction of the sheath, which in turn propels the inner tube along with the spike tip. Tube expulsion facilitates the release of associated effector proteins into target cells or the extracellular space (*Allsopp and Bernal, 2023*; *Coulthurst, 2019*).

Bioinformatic analyses have shown that CISs are conserved across diverse microbial phyla, including diderm and monoderm bacteria, as well as archaea (*Chen et al., 2019*; *Geller et al., 2021*). Based on their distinct modes of action, CIS can be categorized into two main groups: intra-cellular type VI secretion systems (T6SS) and extracellular CIS (eCIS). T6SS are abundant among diderm bacteria and in some archaea, and they are anchored to the cytoplasmic membrane during assembly and firing (*Basler et al., 2012*; *Durand et al., 2015*; *Zachs et al., 2024*). A crucial compo-nent of the T6SS is the baseplate, which serves as a nucleus for T6SS assembly and mediates the binding of T6SS particles to the cytoplasmic membrane until contraction. Upon contraction, the spike and inner tube are propelled out of the cell into an adjacent cell or the medium. eCIS, on the other hand, are assembled as free-floating particles in the cytoplasm and are subsequently released into the extracellular space upon lysis of the producer cell (*Xu et al., 2022*; *Shikuma et al., 2014*; *Hurst et al., 2007*; *Jiang et al., 2019*). Following the release, eCISs attach to the surface of target cells via their tail fibers, followed by contraction and puncturing of the target cell envelope.

Besides T6SS and eCIS, there is accumulating evidence of additional CIS assemblies with distinct modes of action in bacteria. For example, in multicellular cyanobacteria, CIS are anchored to the intra-cellular thylakoid membrane stacks via an extension of the baseplate components. These so-called 'thylakoidCIS' (tCIS) have been proposed to function in cell lysis and the formation of "ghost cells" in response to stress conditions (*Weiss et al., 2022*). Furthermore, several recent studies reported the production of free-floating cytoplasmic CIS particles in the multicellular monoderm *Streptomyces* bacteria, which have been shown to modulate cellular development through presumably slightly different mechanisms (*Casu et al., 2023*; *Vladimirov et al., 2023*; *Nagakubo et al., 2021*; *Nagakubo et al., 2023*; *Nagakubo et al., 2024*).

Notably, many *Streptomyces* genomes contain a class IId eCIS locus, a subtype that is structur-ally less well understood compared to other eCIS subtypes, such as 'antifeeding prophages' (AFPs) from *Serratia*, 'metamorphosis-associated contractile structures' (MACs) from *Pseudoalteromonas luteoviolacea*, AlgoCIS from *A. machipongonensis* or '*Photorhabdus* virulence cassettes' (PVCs) from *P. asymbiotica* (*Lin, 2024*).

*Streptomyces* are best known for producing a wide range of medically and industrially important secondary metabolites, including molecules with antimicrobial, antifungal, anticancer, or immunosup-pressive properties. The production of these molecules is tightly coordinated with the *Streptomyces* developmental life cycle that encompasses functionally distinct filamentous cell types: vegetative hyphae that grow by tip extension and branching to scavenge for nutrients and reproductive (aerial) hyphae that eventually differentiate into chains of spores, which can then be dispersed to restart the life cycle (*Schlimpert and Elliot, 2023*; *Flärdh and Buttner, 2009*). Using *Streptomyces coelicolor* as a model organism, we and others previously demonstrated that *S. coelicolor* CIS (CIS^Sc) mediate a form of regulated cell death, which influences the onset of sporulation and secondary metabolite production in response to exogenous stress and unknown cellular signals (*Figure 1*; *Casu et al., 2023*; *Vladimirov et al., 2023*).

It remains unclear, however, whether CIS^Sc contraction and firing occur while the CIS^Sc particles are free-floating in the cytoplasm or whether this depends on an interaction with the membrane. More-over, CIS encoded in *Streptomyces* genomes lacks clear homologs for tail fiber components or a T6SS trans-envelope complex, raising the question of how CIS^Sc particles could potentially interact with the cytoplasmic membrane. We and others recently noted an uncharacterized protein (*Casu et al., 2023*; *Vladimirov et al., 2023*), SCO4242, which is conserved in the majority of *Streptomyces* species and other Actinomycete species that carry a type IId eCIS locus (*Chen et al., 2019*; *Geller et al., 2021*; *Figure 1—figure supplement 1*). Interestingly, SCO4242 is encoded just downstream of the base-plate components in the CIS^Sc gene cluster, a location that usually encodes tail fibers from conven-tional eCIS (*Xu et al., 2022*; *Jiang et al., 2019*; *Desfosses et al., 2019*). We will, therefore, refer to the gene product of SCO4242 as CisA for 'CIS^Sc-associated protein A'. Here, we set out to assess the importance of CisA for CIS^Sc function and its potential role as a membrane anchor.

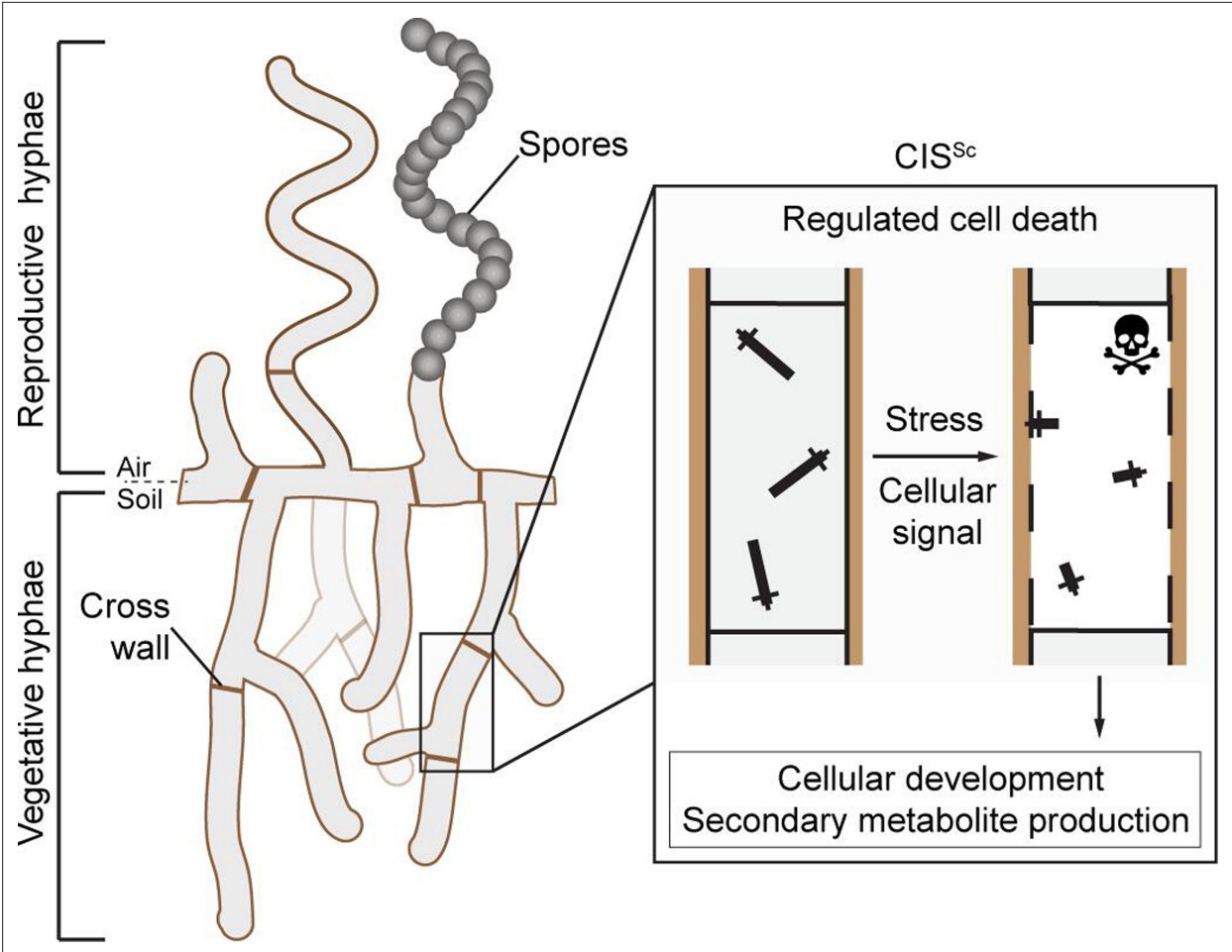

**Figure 1.** Schematic of the mode of action of the intracellular contractile injection system CIS[Sc] in *Streptomyces coelicolor*. CIS[Sc] are assembled as free-floating particles in the cytoplasm of vegetative hyphae (***Casu et al., 2023***; ***Vladimirov et al., 2023***). In response to extracellular stress and/or an unknown cellular signal, CIS[Sc] particles contract, resulting in regulated cell death (mediated by released effectors) and impacting cellular development.

The online version of this article includes the following source data and figure supplement(s) for figure 1:

**Figure supplement 1.** CisA is conserved among CIS-positive *Streptomyces* and actinomycete species.

**Figure supplement 1—source data 1.** Genome accession numbers and BLAST results used to generate ***Figure 1—figure supplement 1***.

## Results

### CisA is required for CIS[Sc] contraction in situ

To explore the role of CisA (SCO4242) in CIS[Sc] contraction and function, we first generated a *S. coelicolor* Δ*cisA* null mutant. Negative stain electron microscopy (EM) imaging of CIS[Sc] particles that were purified from Δ*cisA* mutant cells showed assemblies in the contracted conformation, similar to the CIS[Sc] particles purified from the wild type (WT; ***Figure 2a, b***). Next, to examine the impact of CisA on the behavior of CIS[Sc] in situ, we imaged vegetative hyphae of the WT, the Δ*cisA* mutant, and the complemented mutant (Δ*cisA/cisA*[+]) by cryo-electron tomography (cryoET; 270 tomograms in total, n=3 experiments per strain). We previously demonstrated that CIS[Sc] contraction correlates with the cellular integrity of *S. coelicolor* wild-type (WT) hyphae and showed that in intact hyphae, CIS[Sc] particles are consistently observed in their extended conformation in the cytoplasm, whereas in partially lysed hyphae, the ratio of extended to contracted particles is approximately 2:1, and dead hyphae only contain fully contracted CIS[Sc] assemblies (***Casu et al., 2023***). Interestingly, and in contrast to the WT and the complemented *cisA* mutant strain (***Figure 2c, e***), *cisA*-deficient hyphae only contained CIS[Sc] particles in the extended conformation, irrespective of the cellular integrity of the hyphae (***Figure 2***, ***Figure 2—figure supplement 1***). Moreover, our analysis consistently showed only two CIS[Sc] conformations in intact and dead

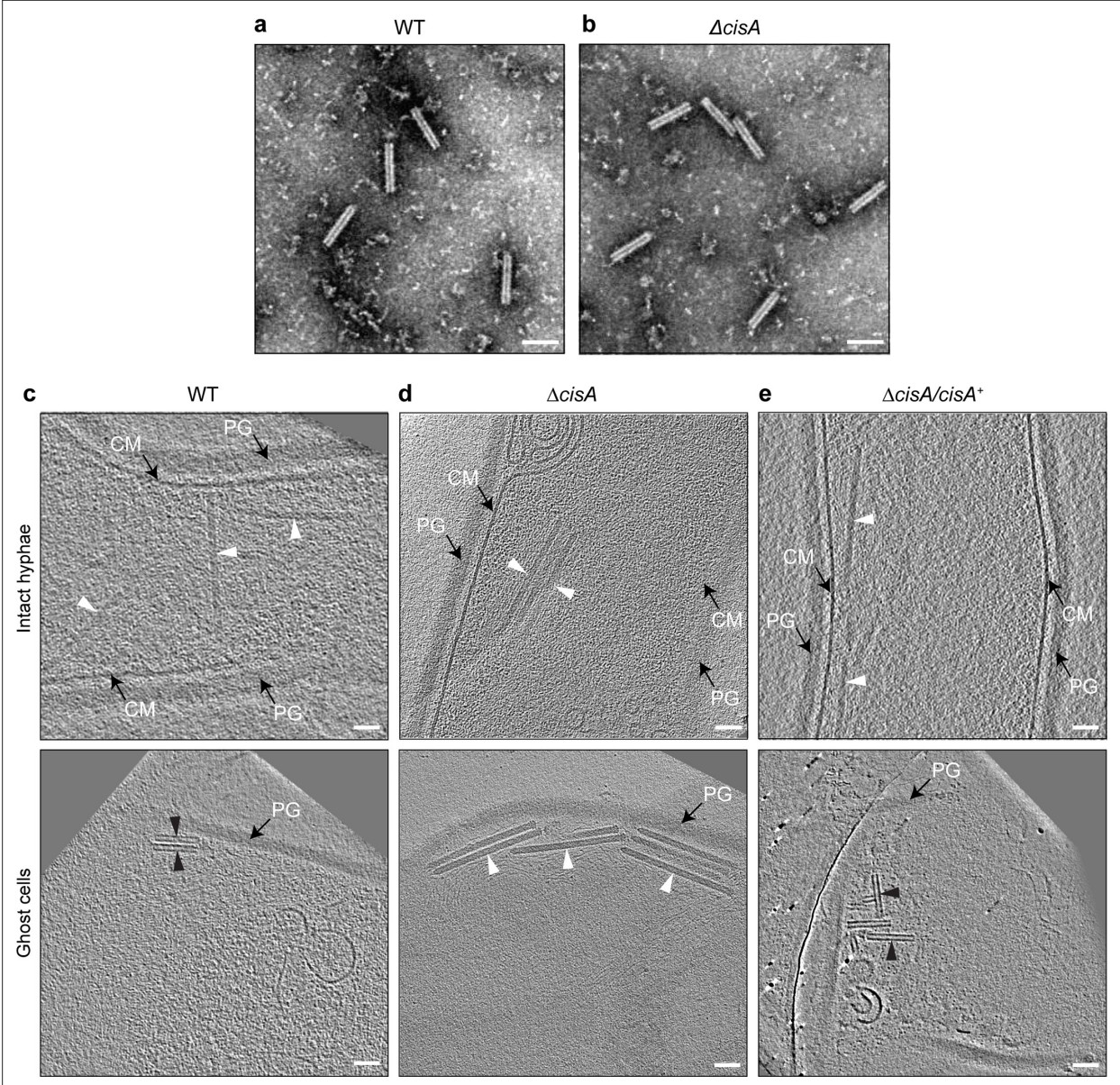

**Figure 2.** Contraction of CIS^Sc in situ is dependent on CisA. (**a, b**) Negative-stain electron micrographs of purified CIS^Sc particles from *S. coelicolor* wildtype (WT) (**a**) and the *ΔcisA* mutant (**b**), show that all CIS^Sc particles are functional and contract upon purification. Experiments were performed three times independently. Bars, 100 nm. (**c–e**) Shown are representative images of cryo-electron tomogram slices (thickness 11 nm) of vegetative hyphae (top: intact cells; bottom: ghost cells) of *S. coelicolor* WT (**c**), *ΔcisA* mutant (**d**), and the complemented *ΔcisA/cisA^+* mutant (**f**). CIS^Sc particles remain almost exclusively in an extended state (white arrowheads) in the *ΔcisA* mutant, whereas in ghost cells derived from the WT and the complemented mutant, CIS^Sc particles (black arrowheads) are mostly contracted. PG, peptidoglycan; CM, cytoplasmic membrane; Bars, 50 nm.

The online version of this article includes the following figure supplement(s) for figure 2:

**Figure supplement 1.** Sheath contraction in situ is linked to the presence of CisA.

hyphae, respectively: the fully extended state (233 nm in length, 18 nm in diameter, n=100 CIS^Sc particles) and the fully contracted state (124 nm in length, 23 nm in diameter, n=100 CIS^Sc particles). While hyphal cell death can be caused by a range of external or internal factors, our results demonstrate that CIS^Sc contraction is not a consequence of cell death and requires CisA in situ.

## CryoEM structure of the extended CIS^Sc assembly

The *S. coelicolor cis* gene cluster consists of 19 predicted open reading frames (accessions *SCO4242-4260*; *Figure 3a*), including genes with sequence similarities to potential CIS structural

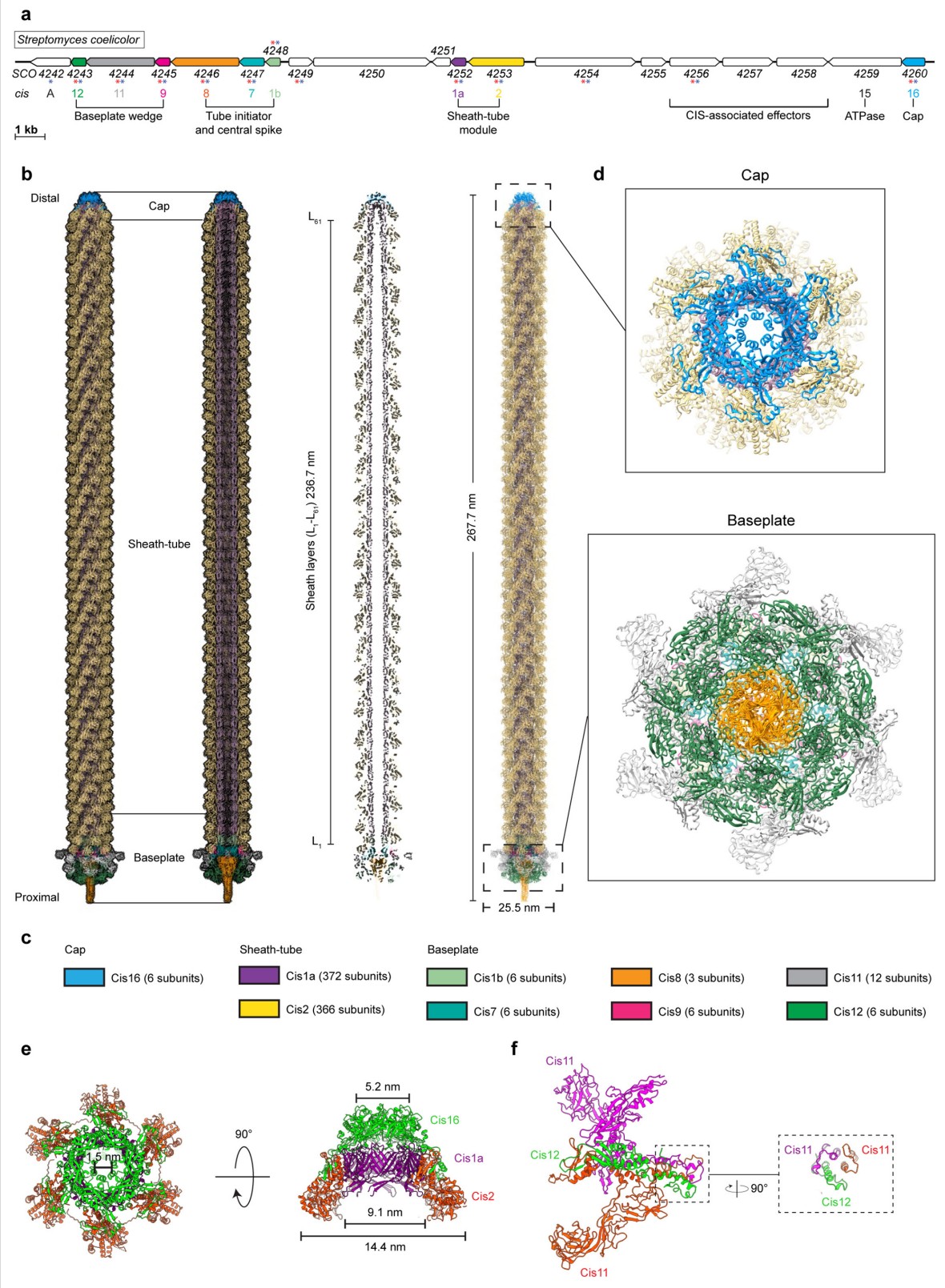

**Figure 3.** CryoEM structure of the extended CIS$^{Sc}$ assembly reveals its composition. (**a**) Schematic illustrating the CIS$^{Sc}$ gene cluster (adapted from Figure 1B from **Casu et al., 2023**). Asterisks indicate gene products that were detected by mass spectrometry analyzing crude preparations of CIS$^{Sc}$ particles (red). See also **Figure 3—source data 1**. (**b, c**) Single-particle cryoEM structure of an extended CIS$^{Sc}$ assembly purified from a non-contractile CIS$^{Sc}$ mutant. Shown are the composite atomic model in surface (left) and ribbon (right) rendering (**b**) Subunits are color-coded according to (**a**), and

*Figure 3 continued on next page*

*Figure 3 continued*

their copy numbers in the assembly are indicated in (**c**). (**d**) Perpendicular views of the ribbon representation of the CIS$^{Sc}$ model of the cap and baseplate modules. (**e**) Top view (left) and side view (right) of ribbon diagrams showing the cap module of CIS$^{Sc}$, revealing that the cap is composed of a Cis16 hexamer. (**f**) Ribbon diagrams showing a single baseplate wedge subunit, which is composed of two copies of Cis11 and one copy of Cis12.

The online version of this article includes the following source data and figure supplement(s) for figure 3:

**Source data 1.** Proteins detected by mass spectrometry in samples of purified CIS$^{Sc}$ sheath particles following treatment with and without nisin.

**Figure supplement 1.** Workflow for the cryoEM structural determination of the extended CIS$^{Sc}$.

**Figure supplement 1—source data 1.** Cryo-EM data statistical analysis.

**Figure supplement 2.** Map qualities of different CIS$^{Sc}$ modules.

**Figure supplement 3.** Structures of the cap modules of other CISs.

**Figure supplement 4.** Structures of the baseplate modules of other CISs.

---

components (*cis1-cis16*), as well as additional genes of unknown functions (including *cisA*) (*Casu et al., 2023*; *Vladimirov et al., 2023*). To determine whether CisA is an integral component of CIS$^{Sc}$, we set out to elucidate the high-resolution structure of purified CIS$^{Sc}$ particles from a non-contractile CIS$^{Sc}$ mutant (*Casu et al., 2023*). The purified assemblies were subjected to single-particle cryo-electron microscopy (cryoEM) for structure determination (*Figure 3—figure supplement 1*), yielding the first high-resolution model of the CIS$^{Sc}$ in its extended conformation, which also presents the first example of a CIS IId subtype (*Chen et al., 2019*).

Our model shows that extended CIS$^{Sc}$ particles share the conserved eCIS architecture, including the three core modules: cap, sheath tube, and baseplate (*Figure 3b*). We processed the dataset using an established workflow that was previously employed to produce the atomic model for other CIS (*Figure 3—figure supplement 1*; *Xu et al., 2022*; *Weiss et al., 2022*). The resulting maps of the baseplate and the cap reached resolutions of 3.5 and 3.4 Å, respectively. The quality of the final maps from the two modules enabled de novo structural modeling (*Figure 3—figure supplement 2*).

The final model of CIS$^{Sc}$ comprises 783 polypeptide chains that assemble into a bullet-shaped particle measuring 268 nm in length (*Figure 3b, c*). All analyzed particles were identical in length, consistent with our in situ data of extended CIS$^{Sc}$ particles in vegetative hyphae of WT *S. coelicolor* (*Figure 2c–e*). The length of eCIS particles is often controlled by tape measure proteins (*Rybakova et al., 2015*). We note, however, that the corresponding gene is absent from the CIS$^{Sc}$ gene cluster. The distal end of the CIS$^{Sc}$ is capped by a tail terminator complex (*Figure 3c, e*). This cap comprises one protein, Cis16, forming a C6-symmetrical complex that terminates the inner tube and sheath, respectively. This minimal architecture resembles the cap complexes of PVC or Pyocin R2 CISs, but it is different from the more complex AlgoCIS from *A. machipongonensis* and AFP from *S. entomophila* (*Figure 3—figure supplement 3*; *Xu et al., 2022*; *Jiang et al., 2019*; *Desfosses et al., 2019*; *Ge et al., 2020*). The structures of the CIS$^{Sc}$ sheath-tube module have been previously discussed, notably showing that the sheath (domains 1/2) contracts by widening and shortening, similar to other CIS (*Casu et al., 2023*). We found that the module consists of 61 sheath layers (Cis2-L$_1$–L$_{61}$) and 60 tube layers (Cis1-L$_1$-L$_{60}$; *Figure 3b, d*). The baseplate module comprises a heteromeric assembly of the proteins Cis1b/7/8/9/11/12 (*Figure 3b, c*). The symmetry transition from the inner tube (Cis1a, C6) to the VgrG-like spike (Cis8, C3) is adapted by two layers of the tube initiators Cis1b and Cis7. The first layer of the sheath (Cis2) is bound to the conserved gp25-like protein Cis9, which connects to the baseplate 'wedges' composed of Cis11 and Cis12 (*Figure 3b, c*). Importantly, the CIS$^{Sc}$ 'wedges' consist of two copies of Cis11 and one copy of Cis12 (*Figure 3f*), resembling the baseplate of Pyocin R2(27). This is in contrast to other previously reported eCIS, whose baseplate wedge comprises only one Cis11/12 component (AlgoCIS, AFP, PVC) and a Cis11 extension forming a cage around the spike (AlgoCIS; *Figure 3—figure supplement 4*; *Xu et al., 2022*; *Jiang et al., 2019*; *Desfosses et al., 2019*). Furthermore, we did not detect any structural components that resemble tail fiber proteins. This is in line with the bioinformatic analysis of the *cis* gene cluster (*Figure 3a*), suggesting the absence of conventional tail fiber genes.

Out of the 19 predicted open-reading frames in the CIS$^{Sc}$ gene cluster (*Figure 3a*), nine encoded proteins were present in the final reconstruction, for which all atomic models were built (*Figure 3b, c*). However, we did not detect a density for CisA in our model, suggesting that CisA is not part of purified CIS$^{Sc}$ particles but may interact with them via a different mechanism. This finding is also in

agreement with mass spectrometry results, which indicate the absence of CisA peptides from purified CIS[Sc] particles (*Figure 3—source data 1*; *Casu et al., 2023*). We speculate that the purification protocol used to isolate CIS[Sc] particles for single-particle analysis may not preserve the CIS[Sc]-CisA interaction or that the interaction is too transient and only occurs under specific growth conditions. To better understand how CisA mediates CIS[Sc] function, we further characterized CisA in vivo.

## CisA is a bitopic protein

Bioinformatic analyses (*Casu et al., 2023*; *Vladimirov et al., 2023*) and structural modeling of CisA suggest that CisA consists of a largely unstructured N-terminal region, a transmembrane segment, and a conserved C-terminal domain (CTD) that shares structural similarity to periplasmic substrate-binding domains (*Figure 4a*). Additional AlphaFold3 (*Abramson et al., 2024*) modeling using just the ordered C-terminus of CisA, including the transmembrane domain and the CTD (amino acids 285–468), predicts that CisA can oligomerize into a pentamer via the CTD (*Figure 4b, c*). These findings suggest that CisA may exist in distinct configurations within the cell. To characterize the intracellular localization of CisA, we first generated a strain in which *cisA* was fused to the N-terminus of *mCherry* and expressed in trans from a constitutive promoter in the *S. coelicolor* Δ*cisA* mutant. Using fluorescence light microscopy, we consistently observed a CisA-mCherry fluorescence signal along the hyphal periphery, indicative of a membrane-associated protein (*Figure 4d*). In contrast, no fluorescence was observed in WT hyphae carrying an empty plasmid (*Figure 4e*). Next, we tested if CisA is indeed localized to the membrane in vivo by performing cellular fractionation experiments. We separated soluble proteins from membrane proteins using whole cell lysates from the *S. coelicolor* WT or the Δ*cisA* mutant that was complemented with a *cisA-3xFLAG* fusion expressed in trans from a constitutive promoter. Our analysis confirmed that CisA-3xFLAG co-sedimented with cell membranes, while the cytoplasmic transcription factor WhiA could only be detected in the soluble protein fraction (*Figure 4f*).

Next, we investigated the topology of CisA in the membrane using a dual *phoA-lacZα* reporter system in *E. coli*. The reporter activity can be directly visualized on indicator plates based on the complementary activities of the cytoplasmic reporter β-galactosidase LacZ (magenta coloration) and the periplasmic reporter alkaline phosphatase PhoA (blue coloration; *Alexeyev and Winkler, 1999*). We engineered *E. coli* strains that expressed a C-terminal fusion of CisA variants to the PhoA-LacZα reporter, including full-length CisA and CisA variants that carried either a deletion in the predicted transmembrane domain and/or the putative periplasmic domain (*Figure 4g*). In addition, we included three controls, including (1) cells carrying the empty reporter plasmid, resulting only in cytoplasmic β-galactosidase activity, (2) cells expressing a PhoA/LacZ fusion to the *Streptomyces* cytoplasmic cell division protein SepH, and (3) cells expressing a PhoA/LacZ fusion to the periplasmic domain of the *Streptomyces* membrane protein RsbN (*Ramos-León et al., 2021*; *Bibb et al., 2012*). The results of this assay confirmed that CisA is a bitopic protein with an N-terminus (amino acids 1–285) located in the cytoplasm and a C-terminus (amino acids 310–468) present in the periplasm of *E. coli* (*Figure 4h*).

Collectively, our results demonstrate that CisA is a membrane protein comprised of a flexible cytoplasmic N-terminal domain and a periplasmic C-terminal domain that is predicted to facilitate CisA oligomerization.

## CisA is essential for the cellular function of CIS[Sc]

We previously established a fluorescence-based assay to show that the production of functional CIS[Sc] particles results in a significantly reduced viability of *Streptomyces* hyphae exposed to exogenous stress, including treatment with the membrane-pore forming antibiotic nisin, UV radiation, or the protonophore carbonyl cyanide 3-chlorophenylhydrazone (CCCP; *Casu et al., 2023*). This viability assay is based on the measurement of the ratio of two fluorescent markers, namely cytoplasmic sfGFP produced by viable intact hyphae and the membrane dye FM5-95 to detect intact as well as partially lysed hyphae. Using this assay, we tested whether CisA was essential for CIS[Sc]-mediated cell death and determined the relative viability of the WT, the Δ*cisA,* and the complemented *cisA* (Δ*cisA/cisA*+) strains expressing cytosolic sfGFP. We compared cells exposed to a sublethal concentration of nisin (1 µg/ml for 90 min) to cells not exposed to antibiotic stress. All three strains were found to express CIS[Sc] assemblies under these growth conditions (*Figure 5—figure supplement 1*).

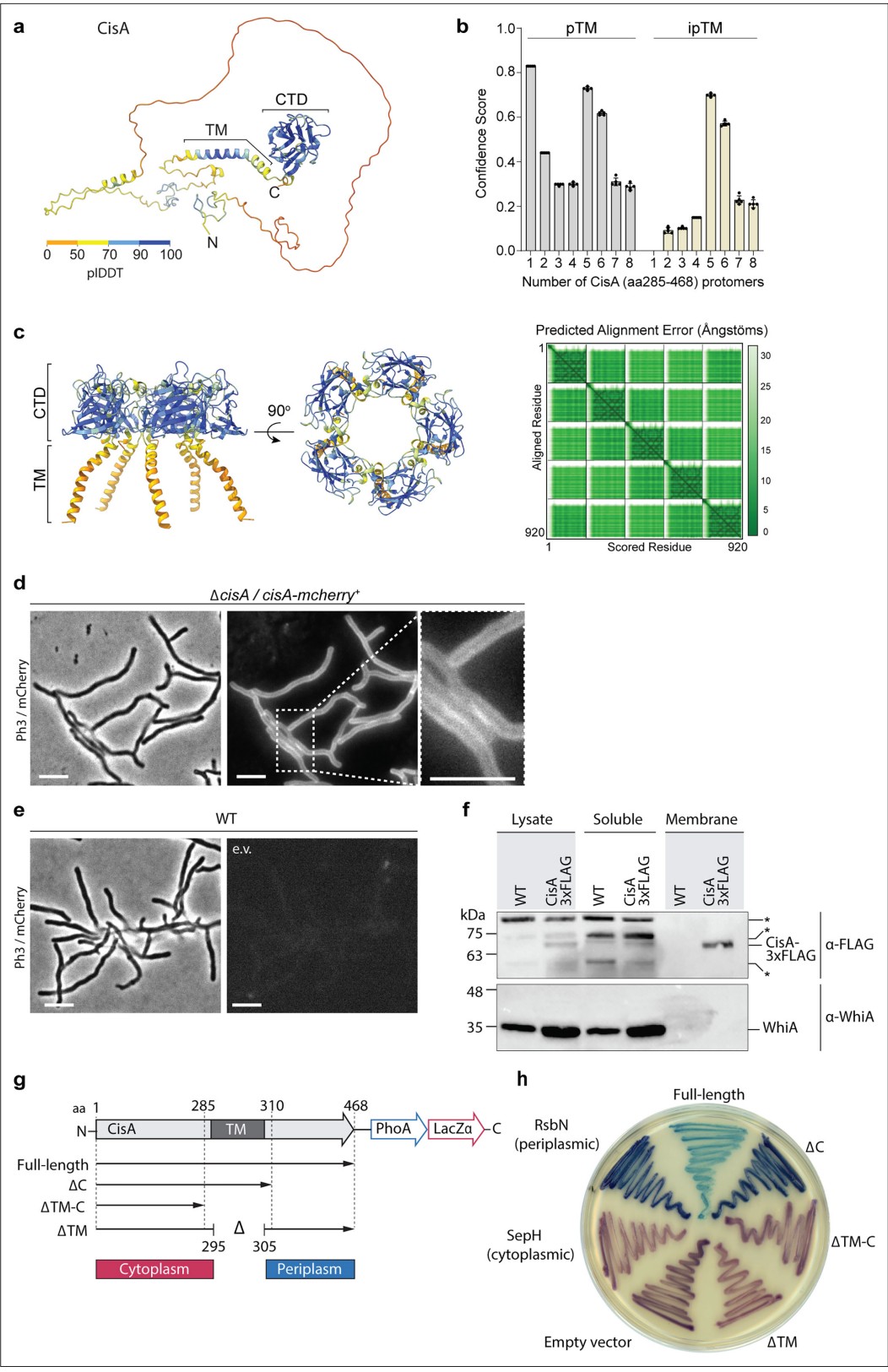

**Figure 4.** CisA is a single-pass membrane protein predicted to oligomerize. (**a**) AlphaFold3 model of CisA colored in pLDDT values. CisA is predicted to contain a largely unstructured N-terminal domain, a transmembrane domain (TM), and a conserved C-terminal domain (CTD). (**b**) Confidence scores for the predicted oligomeric configuration of CisA. Shown are the predicted template modeling (pTM) and the predicted interface pTM scores (ipTM)

*Figure 4 continued on next page*

*Figure 4 continued*

obtained from n=5 predictions per CisA configuration. An ipTM score greater than 0.7 indicates a likely protein–protein interaction. CisA aa 285–468, including the transmembrane domain and the CTD (amino acid 285–468), was used as input sequences for modeling. (**c**) AlphaFold3 model of a CisA (amino acids 285–468) pentamer and the corresponding predicted alignment error (PAE) plot. (**d, e**). Representative micrographs (left panels: phase contrast, Ph3; right panel: mCherry) of strains of *S. coelicolor* vegetative hyphae either constitutively expressing CisA-mCherry (Δ*cisA/cisA-mcherry*) in trans (**d**) or WT cells carrying an empty vector (e.v.) to control for background fluorescence (**e**). Box shows a magnified region of hyphae with CisA-mCherry accumulation in the cellular periphery. Bars, 5 μm. (**f**) Western blot of the cellular fractionation of samples from the *S. coelicolor* WT or the Δ*cisA* mutant constitutively expressing a CisA-3xFLAG fusion in trans. Lysate and soluble and membrane fractions were probed for the presence of CisA-3xFLAG with an α-FLAG antibody (top). Fractionation efficiency was assessed using an α-WhiA antibody to detect the soluble transcription factor WhiA (bottom). The same volumes for each fraction were loaded and analyzed by immunoblotting. CisA-3xFLAG was detected in the lysate and the membrane fraction. Asterisks denote non-specific signals. The experiment was performed in biological duplicates, and shown is a representative image. (**g, h**) Experimental determination of the CisA membrane topology using the dual *phoA-lacZα* reporter in *E. coli*. The schematic diagram (**g**) depicts the four CisA constructs used in the assay and indicates the relevant amino acid (aa) deleted in the three CisA mutant derivatives. The analysis of colony coloration (**h**); pink coloration for cytosolic proteins and blue coloration for periplasmic proteins indicates that the CisA N-terminus is localized in the cytoplasm and the CisA C-terminus is localized to the periplasm, and this topology is dependent on the transmembrane domain (TM). *E. coli* strains carrying an empty reporter plasmid or reporter plasmids with the *Streptomyces* genes for SepH (cytoplasmic) and RsbN (periplasmic) were used as controls. Note that the expression of full-length CisA in *E. coli* is toxic, resulting in reduced growth and a lighter blue coloration.

The online version of this article includes the following source data for figure 4:

**Source data 1.** AlphaFold3 confidence scores used to generate graph in *Figure 4b*.

**Source data 2.** Uncropped micrographs used to generate *Figure 4d, e*.

**Source data 3.** Original files for western blot analysis displayed in *Figure 4f*.

**Source data 4.** Original files of raw immunoblots with relevant bands labeled, related to *Figure 4f*.

Without nisin stress, all three strains exhibited a similar sfGFP/FM5-95 ratio, indicating no significant difference in viability (*Figure 5a, c*). As previously observed (*Casu et al., 2023*), the viability of the WT was dramatically reduced in response to nisin stress. In contrast, the viability of Δ*cisA* hyphae was unaffected by nisin stress and comparable to the viability of untreated cells. This phenotype could be completely reversed by complementing the Δ*cisA* deletion in trans (*Figure 5b, d*). Notably, the Δ*cisA* strain phenocopies CIS[Sc] knockout and non-contractile mutants we analyzed previously (*Casu et al., 2023*). Thus, we conclude that CisA is required for CIS[Sc]-mediated cell death in response to stress.

To further test the dependence of CIS[Sc] function on CisA, we conducted additional in vivo assays. Previous studies showed that the absence of functional CIS[Sc] impacted the timely differentiation of *Streptomyces* hyphae into chains of spores, resulting in accelerated cellular development and reduced secondary metabolite production (*Casu et al., 2023*; *Vladimirov et al., 2023*). Here, we repeated these analyses using the WT, the Δ*cisA* mutant, and the complemented mutant (Δ*cisA/cisA*[+]). All strains consistently completed their life cycle and synthesized spores after 96 hrs of growth on solid medium (*Figure 5e*). Importantly, unlike the WT and the Δ*cisA/cisA*[+] strain, Δ*cisA* mutant colonies sporulated markedly earlier (+ hr). These findings were further corroborated by quantifying the number of spores produced by each strain under the same experimental conditions (*Figure 5f*). Finally, we examined the effect of a *cisA* deletion on secondary metabolite production by quantifying the level of actinorhodin production in *S. coelicolor* (*Malpartida and Hopwood, 1984*). Compared to the WT and the complemented mutant strain (Δ*cisA/cisA*[+]), the Δ*cisA* mutant showed significantly reduced levels of actinorhodin production (*Figure 5—figure supplement 2*). Our data therefore supports the conclusion that the cellular function of CIS[Sc] in *S. coelicolor* depends on CisA, and its absence impacts the timely cellular and chemical differentiation of *S. coelicolor* hyphae.

## Discussion

Here, we identify CisA as an essential membrane-associated factor that mediates the cellular function of CIS[Sc] particles in *Streptomyces coelicolor*. In the absence of CisA, CIS[Sc] particles are unable

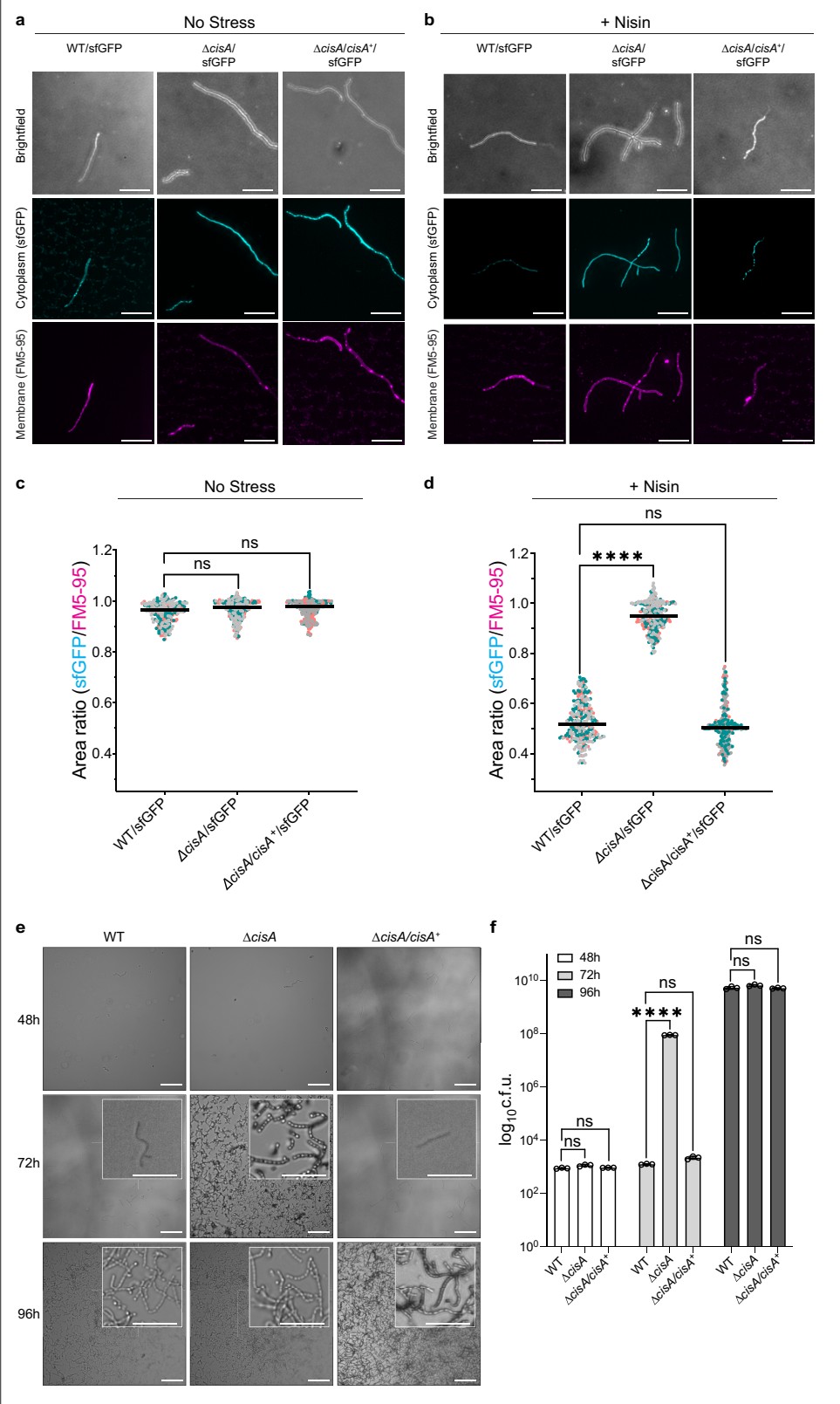

**Figure 5.** CisA is required for regulated cell death and cellular function of CIS[Sc]. (**a-d**) Light microscopy-based viability assay showing that CIS[Sc] does not mediate cell death in *cisA*-deficient hyphae in response to exogenous stress. Shown are representative micrographs of the WT/sfGFP, the Δ*cisA*/sfGFP, and the complemented Δ*cisA*/ *cisA*⁺/sfGFP mutant expressing cytosolic sfGFP (indicator for live cells). Strains were grown for 48 hr and then

*Figure 5 continued on next page*

*Figure 5 continued*

either incubated without (**a**) or with (**b**) the membrane-disrupting antibiotic nisin (1 μg/ml nisin for 90 min). Samples were briefly vortexed to break up mycelial clumps and subsequently stained with the fluorescent membrane dye FM5-95. Bars, 10 μm. Z-stacks were taken of the samples, and the resulting relative fluorescence ratios of sfGFP to FM5-95 are shown in c, d. WT and *ΔcisA/cisA*[+] + cells showed a significantly higher rate of cell death in response to nisin treatment than *ΔcisA* cells. There is no significant difference in the viability of the tested strains in the absence of nisin. Experiments were performed in three biological replicates, as indicated by the green, gray, and magenta data points. Black lines indicate the mean ratio derived from biological triplicate experiments (n=100 images per replicate). ns (not significant) and **** ($p < 0.0001$) were determined using a one-way ANOVA and Tukey's post-test. (**e**) Visualization of spore production in WT, *ΔcisA* mutant, and *ΔcisA/cisA*[+] reveals accelerated cellular development and sporulation of the *ΔcisA* mutant. Shown are representative brightfield images of surface impressions of plate-grown colonies of each strain taken at the indicated time points. Only sporulating hyphae will attach to the hydrophobic surface of the cover glass. Insets show magnified regions of the colony surface containing spores and spore chains. Bars, 50 μm. (**f**) Quantification of sporulation shown in (**e**) via counts of colony-forming units (c.f.u.). The graph shows a sixfold higher c.f.u. count at 72 h in the *ΔcisA* mutant. Strains were grown on R2YE agar, and spores were harvested after 48 hr, 72 hr, and 96 hr of incubation. Data show mean ± s.d. obtained from biological triplicate experiments. ns (not significant) and **** ($p < 0.0001$) were determined using a one-way ANOVA and Tukey's post-test.

The online version of this article includes the following source data and figure supplement(s) for figure 5:

**Source data 1.** Raw data for *Figure 5c, d*.

**Source data 2.** Raw data for *Figure 5f*.

**Figure supplement 1.** Control experiment showing that the behavior of purified CIS[Sc] particles is wild-type-like in the strains used in the viability assay.

**Figure supplement 2.** CisA impacts secondary metabolite production.

**Figure supplement 2—source data 1.** Raw data for *Figure 5—figure supplement 2*.

**Figure supplement 3.** Control experiment showing equal level of cytoplasmic sfGFP in the different strains used for the analysis, displayed in *Figure 5a, b*.

**Figure supplement 3—source data 1.** Original files for western blot analysis shown in *Figure 5—figure supplement 3*.

**Figure supplement 3—source data 2.** Original files of raw immunoblots with relevant bands labeled, related to *Figure 5—figure supplement 3*.

to contract in the cellular context (*Figure 2*). Similar to a CIS[Sc] deletion or non-contractile mutant (*Casu et al., 2023*), the deletion of *cisA* decreases the induction of cell death upon stress, which in turn affects cellular development and secondary metabolite production (*Figure 5*). The mechanistic aspects of this distinct mode of action of CIS[Sc] are discussed below.

A conserved feature of all previously studied CIS is the requirement of the contractile apparatus to bind to a membrane before firing. This can be achieved in various ways, resulting in the classification of CIS into distinct modes of action. First, T6SS are anchored to the cytoplasmic membrane of the host cell by a trans-envelope complex spanning from the cytoplasm to the inner membrane, periplasm, and outer membrane (*Rapisarda et al., 2019*; *Lien et al., 2024*; *Bongiovanni et al., 2024*). Second, contractile phages and eCIS bind to the surface of their target cell via tail fibers, which are typically connected to the baseplate and transmit the signal for firing to the baseplate (*Xu et al., 2022*; *Jiang et al., 2019*; *Ouyang et al., 2024*). Third, tCIS are anchored to thylakoid membrane stacks in cyanobacteria by an extension of their baseplate (*Weiss et al., 2022*). Finally, and in contrast to the previously described CIS, we demonstrate that intracellular CIS[Sc] particles from *Streptomyces* do not contain structural components that can directly bind to the host membrane. Our data suggest that CisA is required to mediate an interaction of CIS[Sc] with the cytoplasmic membrane prior to firing (*Figure 6*). This interaction may occur directly through the binding of CIS[Sc] particles to the membrane protein CisA, or interaction with an as-yet unknown factor. To explore this idea, we performed an in silico prediction of protein-protein interactions between monomeric CisA and CIS[Sc] components using AlphaFold2-Multimer (*McLean, 2024*; *Figure 6—figure supplement 1a*). Interestingly, this analysis identified the baseplate component Cis11 as a significant hit and possible interaction partner of CisA (*Figure 6—figure supplement 1b–e*). Importantly, such a protein-protein interaction would

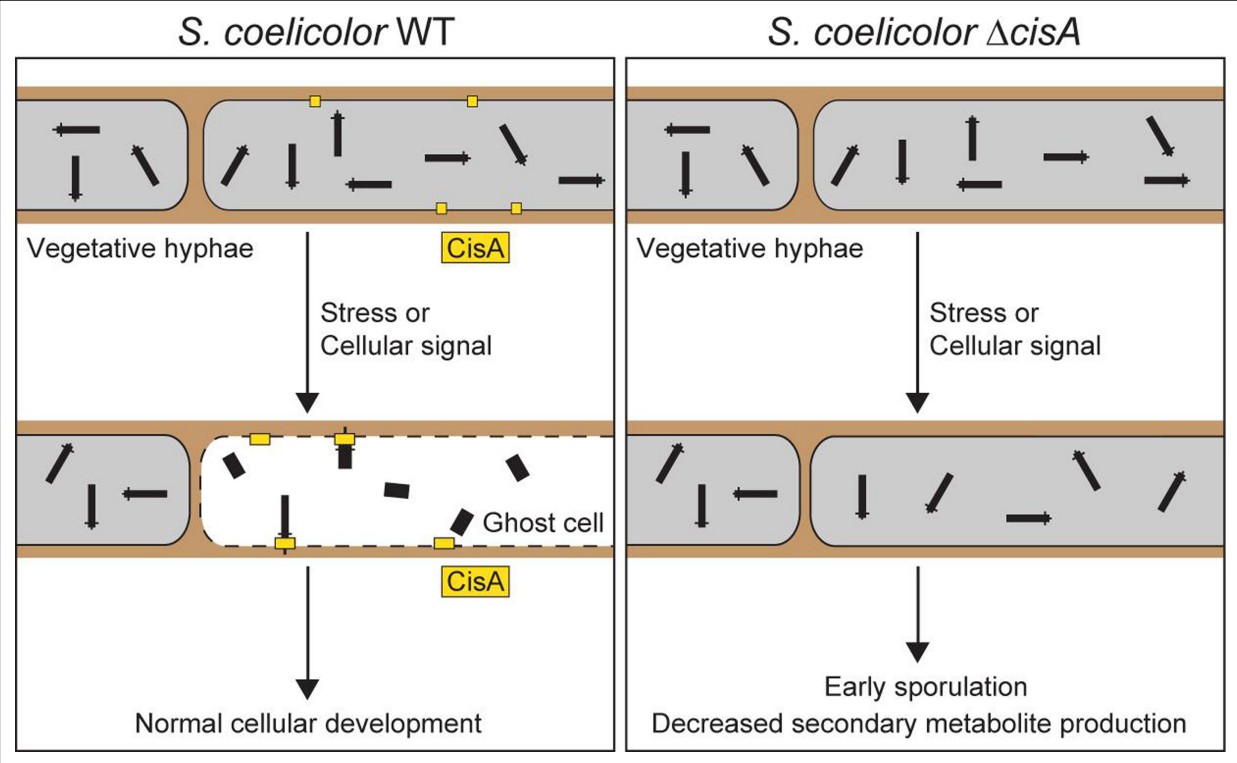

**Figure 6.** Proposed role of the membrane protein CisA in mediating the contraction and firing of cytoplasmic CIS$^{Sc}$ in *Streptomyces coelicolor*. In response to exogenous stress or an unknown cellular signal, membrane-bound CisA either directly or indirectly mediates the association of free-floating CIS$^{Sc}$ particles with the cytoplasmic membrane. This interaction triggers CIS$^{Sc}$ firing and cell death. In hyphae lacking *cisA*, CIS$^{Sc}$ particles mostly remain in their extended (non-contracted) state, leading to reduced hyphal cell death and, consequently, affecting the developmental life cycle of *Streptomyces*.

The online version of this article includes the following source data and figure supplement(s) for figure 6:

**Figure supplement 1.** In silico protein-protein interaction analyses predict that CisA binds a CIS$^{Sc}$ baseplate component.

**Figure supplement 1—source data 1.** LazyAF output with confidence scores for in silico prediction of protein-protein interactions between monomeric CisA and CIS$^{Sc}$ components.

**Figure supplement 1—source data 2.** Structural model shown in *Figure 6—figure supplement 1e*.

**Figure supplement 2.** The cytosolic part of CisA is predicted to interact with Cis11.

**Figure supplement 2—source data 1.** Structural models shown in *Figure 6—figure supplement 2d*.

be consistent with our cryoEM structure (*Figure 3b–f*), which shows a peripheral surface-exposed position of Cis11 in the baseplate complex of extended CIS$^{Sc}$.

Additional structural modeling with truncated versions of CisA in complex with Cis11 further supports the idea that the largely unstructured cytosolic portion of CisA is required to interact with Cis11 (*Figure 6—figure supplement 2*). We have been unable to confirm a direct interaction between CisA and Cis11 because the expression of *cisA* is toxic in *E. coli,* which prevented the experimental analysis of the CisA-Cis11 interaction using co-purification approaches and bacterial two-hybrid studies. While we currently cannot exclude the possibility that the proposed CisA-Cis11 interaction involves an additional unknown factor, we did detect CisA peptides in crude purifications of CIS$^{Sc}$ from nisin-stressed cells (*Figure 3—source data 1*). This would be consistent with a transient and/or short-lived interaction of CIS$^{Sc}$ particles with the membrane and CisA. In agreement with this idea, we rarely observed extended CIS$^{Sc}$ assemblies associated with the cytoplasmic membrane via their baseplate in WT cells (*Figure 2c*).

We further speculate that CisA may not only serve as a mediator for the interaction of CIS$^{Sc}$ with the cytoplasmic membrane but also as a sensor and checkpoint for inducing cell death under specific stress conditions or in response to cellular signals. Notably, protein structure searches with Fold-seek (*van Kempen et al., 2024*) using just the conserved extra-cytoplasmic C-terminal CisA domain

(amino acids 310–468) suggest that this domain shares similarity with solute-binding protein domains. Thus, the C-terminal CisA domain could potentially be involved in directly or indirectly sensing stress signals. Recognition or binding of such a signal by monomeric CisA could trigger a conformational change that results in CisA oligomerization, which in turn could lead to an 'activated' CisA configuration that is primed to mediate the recruitment of CIS$^{Sc}$ assemblies to the cytoplasmic membrane, followed by firing.

It is conceivable that the lethal consequences of CIS$^{Sc}$ firing for the producing cell may have driven the evolution of a stepwise mechanism, involving a checkpoint that prevents self-inflicted cell death by accidental firing. Such a stepwise mechanism would ensure that the CIS$^{Sc}$ remain in a free-floating and safe state within the cytoplasm until a stress signal induces CisA-mediated CIS$^{Sc}$ contraction.

## Materials and methods

### Bacterial strains, plasmids, and oligonucleotides

Bacterial strains, plasmids, and oligonucleotides used in this study are listed in *Supplementary file 1* and *Supplementary file 2*. Plasmids were generated using either standard restriction-ligation or assembled using the Gibson Assembly Master Mix (NEB) or In Vivo Assembly. All plasmids were verified by DNA sequencing. *Escherichia coli* strains were cultured in LB, SOB, or DNA medium. *E. coli* strains TOP10, DH5α, and NEB5α were used to propagate plasmids and cosmids, *E. coli* strain BW25113/pIJ790 for recombineering cosmids and *E. coli* ET12567/pUZ8002 for interspecies conjugation. When required, media was supplemented with antibiotics at the following concentrations: 100 µg/ml carbenicillin, 50 µg/ml apramycin, 50 µg/ml kanamycin, 50 µg/ml hygromycin.

*Streptomyces coelicolor* strains were cultivated in LB, TSB, TSB-YEME, or R2YE liquid medium at 30 °C in baffled flasks or flasks with springs, at 250 rpm or grown on LB, SFM, R2YE medium solidified with 1.5% (w/v) Difco agar (*Chater et al., 2000*). Antibiotics were added at the following concentrations: 25 µg/ml apramycin, 50 µg/ml kanamycin, 25 µg/ml hygromycin, 12.5 µg/ml nalidixic acid.

### Construction of the *S. coelicolor* Δ*cisA* mutant

The $\lambda$ RED homologous recombination system was used to isolate gene replacement mutations using PCR-directed mutagenesis (ReDirect) of the *S. coelicolor* cosmid StD8A containing the *cis* gene cluster (*Gust et al., 2003*; *Gust et al., 2004*). The *cisA* coding sequence (*SCO4242*) was replaced with the *aac3(IV)-oriT* resistance cassette from pIJ773. The mutant cosmid (pSS684) was introduced into *E. coli* ET12567/pUZ8002, followed by conjugation into *S. coelicolor* M145. Exconjugants that had successfully undergone double-homologous recombination were identified by screening for apramycin resistance and kanamycin sensitivity. The deletion of *cisA* was subsequently verified by PCR.

### Sheath preparation of CIS$^{Sc}$

*S. coelicolor* strains were grown in 30 ml TSB, TSB-YEME or R2YE liquid medium for 48 hr. Cells were pelleted by centrifugation (7000 × *g*, 10 min, 4 °C), resuspended in 5 ml lysis buffer (150 mM NaCl, 50 mM Tris-HCl, 0.5×CellLytic B (Sigma-Aldrich), 1% Triton X 100, 200 µg/ml lysozyme, 50 µg/ml DNAse I, pH 7.4), and incubated for 1 hr at 37 °C. Cell debris was removed by centrifugation (15,000 × *g*, 15 min, 4 °C), and cleared lysates were subjected to ultra-centrifugation (150,000 × *g*, 1 hr, 4 °C). Pellets were resuspended in 150 µl resuspension buffer (150 mM NaCl, 50 mM Tris-HCl, supplemented with protease inhibitor cocktail [Roche], pH 7.4). CIS$^{Sc}$ sheath preparations were analyzed by negative stain EM imaging (*Ohi et al., 2004*) and mass spectrometry at the Functional Genomics Center Zürich.

To purify non-contractile CIS$^{Sc}$ particles from *S. coelicolor* (SS393), cleared cell lysates were subjected to ultracentrifugation using a sucrose cushion (20 mM Tris pH 8.0, 150 mM NaCl, 50 mM EDTA, 1% Triton X 100, 50% [w/v] sucrose; *Casu et al., 2023*). The sucrose cushion alongside 1 mm of liquid above and residual bacterial contaminants were removed by centrifugation at 15,000 × *g*. Samples were then subjected to a second round of ultracentrifugation without a sucrose cushion (150,000 × *g*, 1 h, 4 °C), and resulting cell pellets were resuspended in buffer (50 mM Tris, 150 mM NaCl). These crude samples were then further purified by gradient ultracentrifugation (10–50% continuous gradient made with BIOCOMP gradient master IP model 107 gradient forming instrument) at 100,000 × *g* for 1 h using a SW55 Ti rotor. The gradient was analyzed in 11 fractions of 500 µl, which were imaged by

negative stain EM. The fractions, which contained CIS$^{Sc}$ particles, were pooled and used for further experiments.

## Negative stain electron microscopy

4 µl of purified CIS$^{Sc}$ sheath particles were adsorbed to glow-discharged, carbon-coated copper grids (Electron Microscopy Sciences) for 60 s, washed twice with milli-Q water and stained with 2% phosphotungstic acid for 45 s. The grids were imaged at room temperature using a Thermo Fisher Scientific Morgagni transmission electron microscope (TEM) operated at 80 kV.

## Mass spectrometry analysis

To confirm the presence of predicted CIS$^{Sc}$ components, isolated sheath particles were subjected to liquid chromatography–mass spectrometry analysis (LC–MS/MS). These experiments were conducted at the Functional Genomics Center Zürich. First, the samples were digested with 5 µl of trypsin (100 ng/µl in 10 mM HCl) and microwaved for 30 min at 60 °C. The samples were then dried, dissolved in 20 µl ddH$_2$0 with 0.1% formic acid, diluted in 1:10, and transferred to autosampler vials for liquid chromatography with tandem mass spectrometry analysis. A total of 1 µl was injected on a nanoAcquity UPLC coupled to a Q-Exactive mass spectrometer (Thermo Fisher). Database searches were performed by using the Mascot SwissProt and tremble_streptomycetes search programs. For search results, stringent settings have been applied in Scaffold (1% protein false discovery rate, a minimum of two peptides per protein, 0.1% peptide false discovery rate). The results were visualized by Scaffold software (Proteome Software Inc, Version 4.11.1).

## Protein gel electrophoresis and western blotting

Protein samples were boiled at 95 °C for 5 min and separated by a 4–20% Mini-PROTEAN TGX Stain-Free SDS PAGE (Bio-Rad) or 12% Tris-Glycine SDS PAGE (Invitrogen) and, if required, visualized using InstantBlue Coomassie Protein Stain or ReadyBlue Protein Gel Stain (Sigma-Aldrich).

For immunoblot analysis, cells were grown in biological triplicate for 48 hr in TSB and 2 ml aliquots of each culture were pelleted and washed once with 1 x PBS. Cell pellets were resuspended in 0.4 ml of sonication buffer (20 mM Tris pH 8.0, 5 mM EDTA, 1 x EDTA-free protease inhibitors [Sigma-Aldrich]) and subjected to sonication at 4.5 micron amplitude for seven cycles of 15 s on / 15 s off. Samples were centrifuged at 17,000 × $g$ for 15 min at 4 °C. Protein concentration in cleared cell lysate supernatants was determined by Bradford Assay (Bio-Rad). Equivalent total protein concentrations (1 mg/ml) were assayed using the semi-dry western blot transfer. The gel was transferred to a PVDF membrane (Bio-Rad) and probed with the appropriate antibody diluted in TBS-T or using the iBIND buffer system (Thermo Fisher): Rabbit anti-FLAG ([Sigma-Aldrich] F7425, 1:2500), Rabbit anti-WhiA (*Bush et al., 2013*, 1:2500), and Goat Anti-Rabbit-HRP (Abcam Ab6721, 1:5000). To visualize HRP-conjugated antibodies, membranes were incubated with SuperSignal West Femto ECL solution (Thermo Fisher).

## Cellular fractionation

To fractionate membrane and soluble proteins from *S. coelicolor* (WT, SS576), cells were grown for 48 hr in 50 ml TSB/YEME and the entire culture volume was pelleted by centrifugation. Cell pellets were resuspended in 1/10 volume of lysis buffer (0.2 M Tris-HCl, pH 8, 10 mg/mL lysozyme, and 1×EDTA-free protease inhibitors; [Roche]), incubated for 30 min at 37 °C, and then briefly cooled on ice before lysed by sonication (11x15 s on/off at 50% power at 8 microns on ice). Cell debris was removed by centrifugation at 16,000 × $g$ for 20 min and the supernatant was subsequently subjected to two additional rounds of centrifugation. The cleared cell lysate was subjected to ultracentrifugation for 1 hr at 150,000v× $g$ at 4 °C to separate the soluble proteins from membrane proteins. Sedimented membrane proteins were resuspended and washed twice in 1 volume of wash buffer (60 mM Tris-HCl, 200 mM NaCl, pH 8, 0.2 mM EDTA, and 0.2 M sucrose) at 150,000 × $g$ at 4 °C for 1 hr. The wash step was repeated one final time with the wash buffer containing 8 M urea to remove traces of membrane-associated proteins (*Gallego-Parrilla et al., 2024*). The final pellet was dissolved in 1/10 of the initial volume with wash buffer (no urea). Equi-volume amounts of fractions were mixed with 2 x SDS sample buffer and analyzed by immunoblotting.

## Membrane protein topology analysis in *E. coli*

The coding sequence of CisA was inserted in the dual Pho-LacZα reporter plasmid pKTop, which encodes an *E. coli* alkaline phosphatase fragment PhoA$_{22-472}$ fused in frame after the α-fragment of β-galactosidase LacZα$_{4-60}$ (*Alexeyev and Winkler, 1999*). For membrane protein topology, *E. coli* TG1 was transformed with the pKTop and derivatives. Membrane protein topology was assayed by plating the resulting reporter strains on dual-indicator plates containing LB agar supplemented with 80 µg/ml 5-bromo-4-chloro-3-indolyl phosphate disodium salt (X-Phos; Sigma-Aldrich, RES1364C-A101X) and 100 µg/ml 6-chloro-3-indolyl-β-d-galactoside (Red-Gal; Sigma-Aldrich, B6149) as indicators, 1 mM IPTG, and 50 µg/ml kanamycin. A periplasmic or extracellular location of the reporter fusion results in higher alkaline phosphatase activity (blue color), whereas a cytosolic location of the reporter leads to higher β-galactosidase activity (magenta color). Plates were incubated for 2 days at 37 °C and then scanned.

## Vitrified sample preparations

For single-particle cryoEM (SPA), the *S. coelicolor* non-contractile CIS particles were purified as described above and vitrified using a Vitrobot Mark IV (Thermo Fisher Scientific). 4 µl of samples were applied twice on glow-discharged 200 mesh Quantifoil copper grids (R 2/2) which were manually coated with a layer of 1 nm carbon. Grids were blotted for 5.5 s and plunge-frozen in liquid ethane-propane mix (37 %/63 %). Frozen grids were stored in liquid nitrogen until loading into the microscope.

For cryo-electron tomography (cryoET), *Streptomyces* cells were mixed with 10 nm Protein A conjugated colloidal gold particles (1:10 v/v, [Cytodiagnostics]) and 4 µl of the mixture was applied to a glow-discharged holey-carbon copper EM grid (R2/1 or R2/2, [Quantifoil]). The grid was automatically blotted from the backside for 4–6 s in a Mark IV Vitrobot by using a Teflon sheet on the front pad and plunge-frozen in a liquid ethane-propane mixture (37 %/63 %) cooled by a liquid nitrogen bath.

## SPA data collection and image processing

44,925 movies were collected on Titan Krios microscope (Thermo Fisher Scientific), operated at 300 keV equipped with K3 direct electron detector, operating in counting mode, and using a slit width of 20 eV on a GIF-Quantum energy filter (Gatan). The automated data collection was conducted with EPU software (Thermo Fisher Scientific) with a final pixel size of 1.065 Å/pix over 40 frames with a total dose of 50 e$^-$/Å$^2$. The targeted defocus was set between –1.6 and –2.8 µm with 0.2 µm increment.

Movie alignments with dose-weighting were performed in MotionCor2 software (*Zheng et al., 2017*) with a gain reference estimated a posteriori *Afanasyev et al., 2015* followed by standard in-house pipeline of manual inspection and selection of 38,845 micrographs (*Afanasyev, 2023*) for further processing in cryoSPARC package v4.0.3 (*Punjani et al., 2017*). Due to the low concentration of particles and the presence of other contaminants in the sample, automatic particle picking appeared challenging, and we developed a CIS-specific approach for efficient picking of the cap and baseplates. In this approach, CIS$^{Sc}$ particles were manually selected as filaments (start-to-end) on 920 micrographs and used for training a crYOLO (*Wagner et al., 2019*) picking model (*Figure 3—figure supplement 1*). The large size of the training set appeared essential for successful picking; we could not get rid of a large number of false-positive picks, but there were not so many false-negative picks. Then, CIS$^{Sc}$ particles were picked in crYOLO as filaments, followed by the extraction of the ends of the 'filaments' using the star_modif.py script (*Afanasyev, 2023*). Particle coordinates were imported in cryoSPARC, where 211,041 particles were extracted at a large box size of 756 pixels and binned to 64 pixels for initial 2D classification, which allowed us to separate initial particle sets corresponding to the CIS$^{Sc}$ cap (36,569 particles) and CIS$^{Sc}$ baseplate (43,087 particles; *Figure 3—figure supplement 1*).

Two subsets of this dataset were processed separately in cryoSPARC. For the cap subset, a standard cryoSPARC single-particle processing approach was applied, including gradual iterative 2D classification rounds, followed by selection of good classes and particle re-extraction at finer sampling. The main challenge was the proper centering of the particles selected on the end of elongated filament-like objects with repeating patterns (sheath-tube module of CIS), hampering the alignments. Ab initio model generation with applied C6 symmetry was hampered by poor centering of 2D classes due to the elongated nature of the particles (*Figure 3—figure supplement 1*). Homo-refinement with this reference was performed, followed by particle re-extraction and one more round of ab initio model

generation with applied C6 symmetry (*Figure 3—figure supplement 1*). The new 3D reconstruction appeared reliable and was used for further iterative 3D refinements (Homo-refinement and NU-refinements) accompanied by 2D classification and CTF-refinement rounds to exclude particles with wrong Euler angle assignments. The final 3.4 Å resolution map with imposed C6 symmetry resulted from 19,218 particles.

The baseplate subset was processed using the same approach, resulting in a 3D reconstruction with imposed C6 symmetry and a resolution of 3.5 Å from 22,980 particles (*Figure 3—figure supplement 1*). The tip of the baseplate has C3 symmetry; therefore, to solve that part of the map correctly, we exported that particle stack to Relion4 (*Scheres, 2012*) and took advantage of 2D classification in Relion to select 18,124 particles, corresponding to the best 2D class averages. 3D refinement with relaxed symmetry resulted in a 3.8 Å 3D resolution reconstruction with imposed C3 symmetry.

## Cryo-electron tomography

Intact *Streptomyces* cells were imaged by cryoET as described previously (*Weiss et al., 2017*). Images were recorded using a Titan Krios 300 kV microscope (Thermo Fisher Scientific) equipped with a Quantum LS imaging filter operated at a 20 eV slit width and with K3 Summit direct electron detectors (Gatan). Tilt series were collected using a bidirectional tilt scheme from –60 to +60° in 2° increments. Total dose was 130–150 e$^-$/Å$^2$ and defocus was kept at –8 μm. Tilt series were acquired using SerialEM (*Mastronarde, 2005*), drift-corrected using alignframes, and reconstructed using the IMOD program suite (*Kremer et al., 1996*). To enhance contrast, tomograms were deconvolved with a Wiener-like filter 'tom_deconv' (*Tegunov and Cramer, 2019*).

## Structural modeling of the CIS$^{Sc}$ complex

The map qualities of different modules of CIS$^{Sc}$ (cap and baseplate) allowed for de novo structural modeling (*Figure 3—figure supplements 1 and 2*). The atomic models of SCO4243-4248, SCO4252-4253, and SCO4260 were built de novo using COOT (*Emsley et al., 2010*). The resulting models were iteratively refined using RosettaCM (*Song et al., 1993*) and real-space refinement implemented in PHENIX (*Adams et al., 2010*). In cases where protein domains could only be partially modeled, side chains were not assigned. Final model validation was done using MolProbity (*Adams et al., 2010*), and correlation between models and the corresponding maps was estimated using mtriage (*Adams et al., 2010*). To illustrate the complete CIS$^{Sc}$ structure, we generated a composite model by integrating symmetry-related protein subunits into a full model of the CIS$^{Sc}$, based on the consistent CIS$^{Sc}$ length observed both in situ and in vitro (*Figure 3b*).

## Fluorescence microscopy

Fluorescence-based cell viability assays were performed as described previously (*Casu et al., 2023*). Briefly, to produce cytoplasmic sfGFP in *Streptomyces*, the coding sequence for sfGFP was introduced downstream of the constitutive promoter *ermE** on an integrating plasmid (pIJ10257). The plasmid was introduced by conjugation to *S. coelicolor* strains (SS430, SS575, SS576). An equal level of cytoplasmic sfGFP in the different strains shown in *Figure 5, b* was confirmed by western blotting (*Figure 5—figure supplement 3*). *Streptomyces* strains were grown in 30 ml of TSB liquid culture at 30 °C with shaking at 250 rpm for 48 hr. Where appropriate, nisin was added to a final concentration of 1 μg/ml 90 min prior to imaging. One ml aliquots of each culture were vortexed for 1 min to break up mycelial clumps, centrifuged for 5 min at 15,000 x *g*, washed twice with 1x PBS, resuspended in 1 ml of 1x PBS containing 5 μg/ml of the red-fluorescent membrane dye FM5-95 (Invitrogen) and then incubated in the dark at room temperature for 10 min followed by two wash steps with 1 x PBS. Washed cell pellets were resuspended in a total volume of 50 μl PBS, and 10 μl of each sample was spotted onto 1% agarose pads and subsequently imaged using the Thunder imager 3D cell culture microscope (Leica). First, tile scan images were acquired using the Las X Navigator plug-in software (Leica Application Suite X, Version 3.7.4.23463), and 100 regions of interest were picked manually. Then Z-stack images were acquired using a HC PL APO 100 x objective with the following excitation wavelengths: GFP (475 nm) and TRX (555 nm). Images were processed using LasX software to apply thunder processing and maximum projection and FIJI to create segmentation and quantify the live (sfGFP)/total cells (FM5-95) area ratio (*Schindelin*

*et al., 2012*). Statistical analyses were performed on data from biological triplicates (n=100 images for each experiment) using a one-way ANOVA and Tukey's post-test in GraphPad Prism 9 (Version 9.3.1).

For imaging the subcellular localization of the CIS adaptor protein in *Streptomyces*, cells (strain JS69 or *S. coelicolor* M145) were grown for 48 hr in TSB/YEME liquid medium. Two µl of each strain were spotted on 1% agarose pads made with water and subsequently imaged. Images were acquired using a Zeiss Axio Observer Z.1 inverted epifluorescence microscope fitted with a sCMOS camera (Hamamatsu Orca FLASH 4), a Zeiss Colibri 7 LED light source, and a Hamamatsu Orca Flash 4.0v3 sCMOS camera. mCherry fluorescence was detected using an excitation/emission bandwidth of 577–603 nm/614–659 nm. Images were collected using Zen Blue (Zeiss) and analyzed using Fiji (*Schindelin et al., 2012*).

## Cover glass impression of *Streptomyces* spore chains

Spore titers of relevant strains were determined by dilution plating (*Casu et al., 2023*). $10^7$ colony-forming units (CFU) of *S. coelicolor* strains (WT, SS539, SS557) were spread onto R2YE agar plates and grown at 30 °C. Sterile glass coverslips were gently applied to the top surface of each bacterial lawn at 48 hr, 72 hr, and 96 hrs post-inoculation. Coverslips were then mounted onto glass microscope slides and imaged using a 40 x objective on a Leica Thunder Imager 3D Cell Culture. Images were processed using FIJI (*Schindelin et al., 2012*).

## Actinorhodin production assay

*S. coelicolor* strains (WT, SS539, SS557) were inoculated into 30 ml R2YE liquid media at a final concentration of $1.5 \times 10^6$ CFU/ml. Cultures were grown in baffled flasks at 30 °C overnight. Cultures were standardized to an $OD_{450}$ of 0.5 and inoculated in 30 ml of fresh R2YE liquid medium. For the quantification of total actinorhodin production, 480 µl of samples were collected at the indicated time points. 120 µl of 5 M KOH was added, samples were vortexed and centrifuged at 5000 x $g$ for 5 min. The weight of each tube was recorded. A Synergy 2 plate reader (BioTek) was used to measure the absorbance of the supernatant at 640 nm. The absorbance was normalized by the weight of the wet pellet.

## Structure prediction and in silico protein-protein interaction screen

CisA structural models (monomer and oligomers) and the CisA-Cis11 complex were predicted using AlphaFold3 (*Abramson et al., 2024*) and further processed using UCSF Chimera (*Pettersen et al., 2004*) or ChimeraX-1.7.1 (*Goddard et al., 2018*).

A pairwise in silico screen for possible interactions between CisA and components encoded by the *S. coelicolor cis* gene cluster (*SCO4242-SCO4260*) was conducted using the AlphaFold2 Multimer-based LazyAF pipeline (*McLean, 2024*; *Jumper et al., 2021*; *Mirdita et al., 2022*). The confidence metric (ipTM) for the top model from each pairwise interaction was tabulated and visualized using Prism (Version 10.2.2) with ipTM scores >0.7, indicating a possible protein-protein interaction (*O'Reilly et al., 2023*).

## Bioinformatic analysis of CisA homologs

To identify CisA homologs, we performed a reciprocal BLAST search in 120 genomes of *Streptomyces* and other *Actinomycete* species that were previously reported to include a type IId eCIS (*Chen et al., 2019*; *Vladimirov et al., 2023*) using the closely related *cisA* homologue from *Streptomyces albus* J1074 (YP_007743954.1) as a query (72% identical to CisA from *S. coelicolor*). Of these 120 genomes, 75 had a reciprocal hit for CisA (*Figure 1—figure supplement 1—source data 1*). Protein sequences of CisA homologs were aligned using Clustal Omega (*Madeira et al., 2024*) and visualized with JalView (Version 2.11.3.3; *Waterhouse et al., 2009*).

## Acknowledgements

We thank ScopeM for instrument access at ETH Zürich and the Functional Genomics Center Zürich for mass spectrometry support. We also thank Pilhofer Lab members for helpful discussions on this work.

## Additional information

### Funding

| Funder | Grant reference number | Author |
| --- | --- | --- |
| Swiss National Science Foundation | 31003A_179255/310030_212592 | Martin Pilhofer |
| European Research Council | 10.3030/679209 | Martin Pilhofer |
| NOMIS Stiftung | | Martin Pilhofer |
| Royal Society | URF\R1\180075 | Susan Schlimpert |
| Royal Society | URF\R\231009 | Susan Schlimpert |
| Biotechnology and Biological Sciences Research Council | BB/T015349/1 | Susan Schlimpert |
| Biotechnology and Biological Sciences Research Council | BB/X01097X/1 | Susan Schlimpert |

The funders had no role in study design, data collection and interpretation, or the decision to submit the work for publication.

### Author contributions

Bastien Casu, Conceptualization, Data curation, Formal analysis, Supervision, Investigation, Visualization, Methodology, Writing – original draft, Writing – review and editing; Joseph W Sallmen, Formal analysis, Investigation, Methodology, Writing – review and editing; Peter E Haas, Govind Chandra, Pavel Afanasyev, Jingwei Xu, Investigation, Methodology, Writing – review and editing; Martin Pilhofer, Conceptualization, Resources, Formal analysis, Supervision, Funding acquisition, Investigation, Visualization, Methodology, Project administration, Writing – review and editing; Susan Schlimpert, Conceptualization, Resources, Formal analysis, Supervision, Funding acquisition, Investigation, Visualization, Methodology, Writing – original draft, Project administration, Writing – review and editing

### Author ORCIDs

Bastien Casu ⓘ https://orcid.org/0000-0003-3508-6804
Govind Chandra ⓘ https://orcid.org/0000-0002-7882-6676
Pavel Afanasyev ⓘ https://orcid.org/0000-0002-6353-6895
Martin Pilhofer ⓘ https://orcid.org/0000-0002-3649-3340
Susan Schlimpert ⓘ https://orcid.org/0000-0001-6364-8056

Reviewer #2 (Public review): https://doi.org/10.7554/eLife.104064.3.sa1
Reviewer #3 (Public review): https://doi.org/10.7554/eLife.104064.3.sa2
Author response https://doi.org/10.7554/eLife.104064.3.sa3

# Additional files

### Supplementary files

Supplementary file 1. Complete list of strains and plasmids used in this study.

Supplementary file 2. Oligonucleotides used in this study.

MDAR checklist

### Data availability

Representative reconstructed tomograms (EMD-50935, EMD-50939, EMD-50940, EMD-50944, EMD-50948, EMD-50949) and SPA cryoEM maps (EMD-51564, EMD-51565 and EMD-51566) have been deposited in the Electron Microscopy Data Bank. Atomic models (PDB: 9GTP, PDB: 9GTR, PDB: 9GTS) have been deposited in the Protein Data Bank. All data generated or analysed during this study are

included in the manuscript and supporting files; source data files have been provided for *Figures 1, 3–6*.

The following datasets were generated:

| Author(s) | Year | Dataset title | Dataset URL | Database and Identifier |
|---|---|---|---|---|
| Casu B, Sallmen JW, Hass PE, Afanasyev P, Xu J, Schlimpert S, Pilhofer M | 2025 | Cryo electron tomogram of Streptomyces coelicolor - ScoDelta4242 membrane protein CisA mutant - Intact hyphae | https://www.emdataresource.org/EMD-50935 | EMDataResource, EMD-50935 |
| Casu B, sallmen JW, Hass PE, Afanasyev P, Xu J, Schlimpert S, Pilhofer M | 2025 | Cryo electron tomogram of Streptomyces coelicolor - ScoDelta4242 membrane protein CisA mutant - Ghost cells | https://www.emdataresource.org/EMD-50939 | EMDataResource, EMD-50939 |
| Casu B, Sallmen JW, Hass PE, Afanasyev P, Xu J, Schlimpert S, Pilhofer M | 2025 | Cryo electron tomogram of Streptomyces coelicolor - ScoDelta4242 complemented membrane protein CisA strain - Intact hyphae | https://www.emdataresource.org/EMD-50940 | EMDataResource, EMD-50940 |
| Casu B, Sallmen JW, Hass PE, Afanasyev P, Xu J, Schlimpert S, Pilhofer M | 2025 | Cryo electron tomogram of Streptomyces coelicolor - ScoDelta4242 complemented membrane protein CisA strain - Ghost cells | https://www.emdataresource.org/EMD-50944 | EMDataResource, EMD-50944 |
| Casu B, Sallmen JW, Hass PE, Afanasyev P, Xu J, Schlimpert S, Pilhofer M | 2025 | Cryo electron tomogram of Streptomyces coelicolor - Intact hyphae | https://www.emdataresource.org/EMD-50948 | EMDataResource, EMD-50948 |
| Casu B, Sallmen JW, Hass PE, Afanasyev P, Xu J, Schlimpert S, Pilhofer M | 2025 | Cryo electron tomogram of Streptomyces coelicolor - Ghost cells | https://www.emdataresource.org/EMD-50949 | EMDataResource, EMD-50949 |
| Casu B, Sallmen JW, Haas PE, Afanasyev P, Xu J, Schlimpert J, Pilhofer M | 2025 | Cryo-EM structure of a contractile injection system in Streptomyces coelicolor the baseplate complex in extended state applied 6-fold symmetry | https://www.emdataresource.org/EMD-51564 | EMDataResource, EMD-51564 |
| Casu B, Sallmen JW, Hass PE, Afanasyev P, Xu J, Schlimpert S, Pilhofer M | 2025 | Cryo-EM structure of a contractile injection system in Streptomyces coelicolor the baseplate complex in extended state applied 3-fold symmetry | https://www.emdataresource.org/EMD-51565 | EMDataResource, EMD-51565 |
| Casu B, Sallmen JW, Hase PE, Afanasyev P, Xu J, Schlimpert S, Pilhofer M | 2025 | Cryo-EM structure of a contractile injection system in Streptomyces coelicolor the cap portion in extended state | https://www.emdataresource.org/EMD-51566 | EMDataResource, EMD-51566 |
| Casu B, Sallmen JW, Haas PE, Afanasyev P, Xu J, Schlimpert S, Pilhofer M | 2025 | Cryo-EM structure of a contractile injection system in Streptomyces coelicolor, the baseplate complex in extended state applied 6-fold symmetry | https://doi.org/10.2210/pdb9gtp/pdb | Worldwide Protein Data Bank, 10.2210/pdb9gtp/pdb |

*Continued on next page*

*Continued*

| Author(s) | Year | Dataset title | Dataset URL | Database and Identifier |
|---|---|---|---|---|
| Casu B, Sallmen JW, Hass PE, Afanasyev P, Xu J, Schlimpert S, Pilhofer M | 2025 | Cryo-EM structure of a contractile injection system in Streptomyces coelicolor, the baseplate complex in extended state applied 3-fold symmetry | https://doi.org/10.2210/pdb9GTR/pdb | Worldwide Protein Data Bank, 10.2210/pdb9GTR/pdb |
| Casu B, Sallmen JW, Hass PE, Afanasyev P, Xu J, Schlimpert S, Pilhofer M | 2025 | Cryo-EM structure of a contractile injection system in Streptomyces coelicolor, the cap portion in extended state | https://doi.org/10.2210/pdb9gts/pdb | Worldwide Protein Data Bank, 10.2210/pdb9gts/pdb |

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
