## [Editor Report · eLife Assessment]

This **valuable** study provides insights into the structure and function of bacterial contractile injection systems that are present in the cytoplasm of many Streptomyces strains. A **convincing** high-resolution model of the structure of extended forms of the cytoplasmic contractile injection system assembly from Streptomyces coelicolor is presented, with some investigation of the membrane protein CisA in attachment of the extended assembly to the inner face of the cytoplasmic membrane and the firing of the system. The work expands the current understanding of these diverse bacterial nanomachines.

---

## [Referee Report · Reviewer #2 (Public review)]

Summary:

The paper addresses how the S. coelicolor contractile injection system (CISSc) interacts with the membrane, how it contracts and fires, and how it affects both cell viability and differentiation, which it has been implicated to do in previous work from this group and others. The Streptomyces CIS systems have been enigmatic in the sense that they are free-floating in the cytoplasm in an extended form and are seen in contracted conformation (i.e. after having been triggered) mainly in dead and partially lysed cells, suggesting involvement in some kind of regulated cell death. So, how do the structure and function of the CISSc system compare to other types of CIS from other bacteria and phages, does it interact with the cytoplasmic membrane, how does it do that, and is the membrane interaction involved in the suggested role in stress-induced, regulated cell death? The authors address these questions by investigating the role of a membrane protein, CisA, that is encoded by a gene in the CIS gene cluster in S. coelicolor. Further, they show for the first time the structure of the assembled CISSc, purified from the cytoplasm of S. coelicolor, analysed using single-particle cryo-electron microscopy.

Strengths:

The beautiful visualisation of the CIS system both by cryo-electron tomography of intact bacterial cells and by single-particle electron microscopy of purified CIS assemblies are clearly the strengths of the paper, both in terms of methods and results. Further, the paper provides genetic evidence that the membrane protein CisA is required for the contraction of the CISSc assemblies that are seen in partially lysed or ghost cells of the wild type. The conclusion that CisA is a transmembrane protein and the inferred membrane topology are well supported by experimental data. The cryo-EM data suggest that CisA is not a stable part of the extended form of the CISSc assemblies. These findings raise the question of what CisA does. Interestingly, Alphafold modelling suggests that the cytoplasmic part of CisA interacts directly with the base plate protein Cis11.

Weaknesses:

The investigations of the role of CisA in function, membrane interaction, and triggering of contraction of CIS assemblies are key parts of the paper and are highlighted in the title. However, the data presented to answer these questions are partially incomplete and have some limitations.

As an example, although the modelling that suggests interaction between CisA and the base plate protein Cis11 appears compelling, the interaction has not yet been possible to test and verify experimentally. Further, it remains unclear whether or how CisA recruits the CISSc system to the membrane. Overall, the mechanism by which CisA may act on CISSc and cause firing remains largely unclear.

Further, the paper does not provide new insights into the role of the CISSc system in growth or developmental biology of streptomycetes. The assay of how CisA affects the function of the system involves monitoring stress-induced loss of viability based on loss of cytoplasmic GFP signal, as described in a previous paper. The assay looks only at single hyphal fragments released from mycelial networks or mycelial pellets, and it could have been interesting to observe effects also under other growth conditions. Similarly, the effect on the developmental life cycle is limited to showing accelerated sporulation in the CisA mutant, similar to what was previously shown for mutants lacking other parts of the system. The paper shows that CisA is needed for the observed phenotypic effects of the CISSc system, but the overall biological roles of the CISSc and CisA remain elusive.

Concluding remarks:

This paper provides new insights into the structure of the unusual subclass of bacterial contractile injection systems (CIS) that is constituted by the cytoplasmically located systems found in streptomycetes. Importantly, the work also describes a membrane protein, CisA, that likely links the CISSc to the cytoplasmic membrane and is required for its function and likely its triggering. The paper will be of large interest in the field, and it will likely be the basis for further and more mechanistic and functional investigations of the Streptomyces CIS systems.

---

## [Referee Report · Reviewer #3 (Public review)]

Summary

In this work, Casu et al. have reported the characterization of a previously uncharacterized membrane protein CisA encoded in a non-canonical contractile injection system of Streptomyces coelicolor, CISSc, which is a cytosolic CISs significantly distinct from both intracellular membrane-anchored T6SSs and extracellular CISs. The authors have presented the first high-resolution structure of the extended CISSc structure. It revealed important structural insights of the extended state of this non-canonical CIS.

To further explore how CISSc interacted with cytoplasmic membrane, they further set out to investigate a membrane protein CisA encoded in the CISSc cluster and previously hypothesized to be the membrane adaptor for CISSc; however, the structure revealed that it was not associated with CISSc. Using a fluorescence microscope and cell fractionation assay, the authors verified that CisA is indeed a membrane-associated protein. They further determined experimentally that CisA had a cytosolic N-terminal domain and a periplasmic C-terminus. The functional analysis of cisA mutant revealed that it is not required for CISSc assembly but is essential for the contraction, as a result, the deletion significantly affects CISSc-mediated cell death upon stress, timely differentiation, as well as secondary metabolite production. Although the work did not resolve the mechanistic detail how CisA interacts with CISSc structure, they used in-silico prediction of protein-protein interactions between monomeric CisA and CISSc components using Alphafold2-Multimer, which identified baseplate protein Cis11 as a potential interaction partner. Such prediction sets out a strong basis for future investigations to explore the molecular mechanistic details how CisA mediates the contraction via interactions with the CIS structural components such as Cis11. Using AlphaFold3, the authors also estimated the oligomerization state of CisA, which can be present as a pentamer. Authors further suggested that such oligomerization is mediated by the interaction of C-terminal solute-binding like domain.

In general, the work provides solid data and a strong foundation for future investigation toward understanding the mechanism of CISSc contraction, and potentially, the relation between the membrane association of CISSc, the sheath contraction and the cell death.

Major Strength:

The paper is well-structured, and the conclusion of the study is supported by solid data and careful data interpretation were presented. The authors provided strong evidence on (1) the high-resolution structure of extended CISSc determined by cryo-EM, and the subsequent comparison with known eCIS structures, which sheds light on both its similarity and different features from other subtypes of eCISs in detail; (2) the topological features of CisA using fluorescence microscopic analysis, cell fractionation and PhoA-LacZα reporter assays, (3) functions of CisA in CISSc-mediated cell death and secondary metabolite production, likely via the regulation of sheath contraction, (4) structural prediction of the oligomerization state of CisA and potential interaction partners of CIS structure.

Weakness:

Due to technical limitations, authors are not able to experimentally demonstrate the direct interaction between CisA with baseplate complex of CISSc, since they could not express cisA in *E. coli* due to its potential toxicity. Therefore, there is a lack of biochemical analysis of direct interaction between CisA and baseplate wedge. However, they have provided solid AlphaFold2-multimer prediction data and identified baseplate protein Cis11 as a potential interaction partner. Such predictions will guide future work towards biochemical analysis to verify such interaction.

While there is no direct evidence showing that CisA is responsible for tethering CISSc to the membrane upon stress, and the spatial and temporal relation between membrane association and contraction remains unclear, I recognize that this is above the scope of the current work, so I would expect further investigation to address these questions in future.

Conclusion

Overall, the work provides a valuable contribution to our understanding on the structure of a much less understood subtype of CISs, which is unique compared to both membrane-anchored T6SSs and host-membrane targeting eCISs. Authors have successfully demonstrated the role of CisA in the contraction of CISSc, along with solid and detailed analysis of the contraction state of the particles with or without CisA using cryo-ET. Using structural modeling, authors also identified the potential oligomerization state and possible interaction partner within the CIS particle.

Importantly, the work serves as a strong foundation to further investigate how the sheath contraction works here. The work contributes to expanding our understanding of the diverse CIS superfamilies, with significant novelty.

---

## [Author Response]

The following is the authors’ response to the original reviews

**Public Reviews:**

**Reviewer #1 (Public review):**
Contractile Injection Systems (CIS) are versatile machines that can form pores in membranes or deliver effectors. They can act extra or intracellularly. When intracellular they are positioned to face the exterior of the cell and hence should be anchored to the cell envelope. The authors previously reported the characterization of a CIS in Streptomyces coelicolor, including significant information on the architecture of the apparatus. However, how the tubular structure is attached to the envelope was not investigated. Here they provide a wealth of evidence to demonstrate that a specific gene within the CIS gene cluster, cisA, encodes a membrane protein that anchors the CIS to the envelope. More specifically, they show that:- CisA is not required for assembly of the structure but is important for proper contraction and CIS-mediated cell death- CisA is associated to the membrane (fluorescence microscopy, cell fractionation) through a transmembrane segment (lacZ-phoA topology fusions in *E. coli*)- Structural prediction of interaction between CisA and a CIS baseplate component- In addition they provide a high-resolution model structure of the >750-polypeptide Streptomyces CIS in its extended conformation, revealing new details of this fascinating machine, notably in the baseplate and cap complexes.All the experiments are well controlled including trans-complemented of all tested phenotypes.One important information we miss is the oligomeric state of CisA.

Thank you for this suggestion. We now provide information on the potential oligomeric state of CisA. We performed further AlphaFold3 modelling of CisA using an increasing number of CisA protomers (1 to 8). We ran predictions for the configuration using the sequence of the well-folded C-terminal CisA domain (amino acids 285-468), which includes the transmembrane domain and the conserved domain that shares similarities to carbohydrate-degrading domains. The obtained confidence scores (mean values for pTM=0.73, ipTM=0.7, n=5) indicate that CisA can assemble into a pentamer and that this oligomerization is mediated through the interaction of the C-terminal solute-binding like superfamily domain.

We have added this information to the revised manuscript (Fig. 3b/c) and further discuss the possible implications of CisA oligomerization for its proposed mode of action.

While it would have been great to test the interaction between CisA and Cis11, to perform cryo-electron microscopy assays of detergent-extracted CIS structures to maintain the interaction with CisA, I believe that the toxicity of CisA upon overexpression or upon expression in *E. coli* render these studies difficult and will require a significant amount of time and optimization to be performed. It is worth mentioning that this study is of significant novelty in the CIS field because, except for Type VI secretion systems, very few membrane proteins or complexes responsible for CIS attachment have been identified and studied.

We thank this reviewer for their highly supportive and positive comments on our manuscript and we are grateful for their recognition of the novelty of our study, particularly in the context of membrane proteins and complexes involved in CIS attachment.

We agree that further experimental evidence on direct interaction between CisA and Cis11 would have strengthened our model on CisA function. However, as noted by this reviewer, this additional work is technically challenging and currently beyond the scope of this study.

**Reviewer #2 (Public review):**
Summary:The overall question that is addressed in this study is how the S. coelicolor contractile injection system (CISSc) works and affects both cell viability and differentiation, which it has been implicated to do in previous work from this group and others. The CISSc system has been enigmatic in the sense that it is free-floating in the cytoplasm in an extended form and is seen in contracted conformation (i.e. after having been triggered) mainly in dead and partially lysed cells, suggesting involvement in some kind of regulated cell death. So, how do the structure and function of the CISSc system compare to those of related CIS from other bacteria, does it interact with the cytoplasmic membrane, how does it do that, and is the membrane interaction involved in the suggested role in stress-induced, regulated cell death? The authors address these questions by investigating the role of a membrane protein, CisA, that is encoded by a gene in the CIS gene cluster in S. coelicolor. Further, they analyse the structure of the assembled CISSc, purified from the cytoplasm of S. coelicolor, using single-particle cryo-electron microscopy.Strengths:The beautiful visualisation of the CIS system both by cryo-electron tomography of intact bacterial cells and by single-particle electron microscopy of purified CIS assemblies are clearly the strengths of the paper, both in terms of methods and results. Further, the paper provides genetic evidence that the membrane protein CisA is required for the contraction of the CISSc assemblies that are seen in partially lysed or ghost cells of the wild type. The conclusion that CisA is a transmembrane protein and the inferred membrane topology are well supported by experimental data. The cryo-EM data suggest that CisA is not a stable part of the extended form of the CISSc assemblies. These findings raise the question of what CisA does.

We thank Reviewer #2 for the overall positive evaluation of our manuscript and the constructive criticism.

Weaknesses:The investigations of the role of CisA in function, membrane interaction, and triggering of contraction of CIS assemblies, are important parts of the paper and are highlighted in the title. However, the experimental data provided to answer these questions appear partially incomplete and not as conclusive as one would expect.

We acknowledge that some aspects of our work remain unanswered. We are currently unable to conduct additional experiments because the two leading postdoctoral researchers on this project have moved on to new positions. We currently don’t have the extra manpower with a similar skill set to pick up the project.

The stress-induced loss of viability is only monitored with one method: an in vivo assay where cytoplasmic sfGFP signal is compared to FM5-95 membrane stain. Addition of a sublethal level of nisin lead to loss of sfGFP signal in individual hyphae in the WT, but not in the cisA mutant (similarly to what was previously reported for a CIS-negative mutant). Technically, this experiment and the example images that are shown give rise to some concern. Only individual hyphal fragments are shown that do not look like healthy and growing S. coelicolor hyphae. Under the stated growth conditions, S. coelicolor strains would normally have grown as dense hyphal pellets. It is therefore surprising that only these unbranched hyphal fragments are shown in Fig. 4ab.

We thank this Reviewer for their thoughtful criticism regarding the viability assays and the data presented in Figure 4. We acknowledge the importance of ensuring that the presented images reflect the physiological state of *S. coelicolor* under the stated growth conditions and recognize that hyphal fragments shown in Figure 4 do not fully capture the typical morphology of *S. coelicolor.* As pointed out by this reviewer, *S. coelicolor* grows in large hyphal clumps when cultured in liquid media, making the quantification of fluorescence intensities in hyphae expressing cytoplasmic GFP or stained with the membrane dye FM5-95 particularly challenging. To improve the image analysis and quantification of GFP and FM5-95-fluorescent intensities across the three *S. coelicolor* strains (wildtype, *cisA* deletion mutant and the complemented *cisA* mutant), we vortexed the cell samples before imaging to break up hyphal clumps, increasing hyphal fragments. The hyphae shown in our images were selected as representative examples across three biological replicates.

Further, S. coelicolor would likely be in a stationary phase when grown 48 h in the rich medium that is stated, giving rise to concern about the physiological state of the hyphae that were used for the viability assay. It would be valuable to know whether actively growing mycelium is affected in the same way by the nisin treatment, and also whether the cell death effect could be detected by other methods.

The reasoning behind growing *S. coelicolor* for 48 h before performing the fluorescence-based viability assay was that we (DOI: 10.1038/s41564-023-01341-x) and others (e.g.: DOI: 10.1038/s41467-023-37087-7) previously showed that the levels of CIS particles peak at the transition from vegetative to reproductive/stationary growth, thus indicating that CIS activity is highest during this growth stage. The obtained results in this manuscript are consistent with previous results, in which we showed a similar effect on the viability of wildtype versus *cis*-deficient *S. coelicolor* strains (DOI: 10.1038/s41564-023-01341-x) using nisin, the protonophore CCCP and UV radiation. The results presented in this study and our previous study are based on biological triplicate experiments and appropriate controls. Furthermore, our results are in agreement with the findings reported in a complementary study by Vladimirov et al. (DOI: 10.1038/s41467-023-37087-7) that used a different approach (SYTO9/PI staining of hyphal pellets) to demonstrate that CIS-deficient mutants exhibit decreased hyphal death.

Taken together, we believe that the results obtained from our fluorescence-based viability assay provide strong experimental evidence that functional CIS mediate hyphal cell death in response to exogenous stress.

The model presented in Fig. 5 suggests that stress leads to a CisA-dependent attachment of CIS assemblies to the cytoplasmic membrane, and then triggering of contraction, leading to cell death. This model makes testable predictions that have not been challenged experimentally. Given that sublethal doses of nisin seem to trigger cell death, there appear to be possibilities to monitor whether activation of the system (via CisA?) indeed leads to at least temporally increased interaction of CIS with the membrane.

We thank this reviewer for their suggestions on how to test our model further. This is a challenging experiment because we do not know the exact dynamics of how nisin stress is perceived and transmitted to CisA and CIS particles.

In an attempt to address this point, we have performed co-immunoprecipitation experiments using *S. coelicolor* cells that produced CisA-FLAG as bait, and which were treated with a sub-lethal nisin concentration for 0/15/45 min. Mass spectrometry analysis of co-eluted peptides did not show the presence of CIS-associated peptides at the analyzed timepoints. While we cannot exclude the possibility that our experimental assay requires further optimization to successfully demonstrate a CisA-CIS interaction (e.g. optimization of the use of detergents to improve the solubilization of CisA from *Streptomyces* membrane, which is currently not an established method), an alternative and equally valid hypothesis is that the interaction between CIS particles and CisA is transient and therefore difficult to capture. We would like to mention, however, that we did detect CisA peptides in crude purifications of CIS particles from nisin-stressed cells (Supplementary Table 2, manuscript: line 301/302), supporting our proposed model that CisA can associate with CIS particles in vivo.

Further, would not the model predict that stress leads to an increased number of contracted CIS assemblies in the cytoplasm? No clear difference in length of the isolated assemblies if Fig. S7 is seen between untreated and nisin-exposed cells, and also no difference between assemblies from WT and cisA mutant hyphae.

The reviewer is correct that there is no clear difference in length in the isolated CIS particles shown in Figure S7. This is in line with our results, which show that CisA is not required for the correct assembly of CIS particles and their ability to contract in the presence and absence of nisin treatment. The purpose of Figure S7 was to support this statement. We would like to note that the particles shown in Figure S7 were purified from cell lysates using a crude sheath preparation protocol, during which CIS particles generally contract irrespective of the presence or absence of CisA. Thus, we cannot comment on whether there is an increased number of contracted CIS assemblies in the cytoplasm of nisin-exposed cells. To answer this point, we would need to acquire additional cryo-electron tomograms (cyroET) of the different strains treated with nisin. CryoET is an extremely time and labor-intensive task and given that we currently don’t know the exact dynamics of the CIS-CisA interaction following exogenous stress, we believe this experiment is beyond the scope of this work.

The interaction of CisA with the CIS assembly is critical for the model but is only supported by Alphafold modelling, predicting interaction between cytoplasmic parts of CisA and Cis11 protein in the baseplate wedge. An experimental demonstration of this interaction would have strengthened the conclusions.

We agree that direct experimental evidence of this interaction would have further strengthened the conclusions of our study, and we have extensively tried to provide additional experimental evidence. Unfortunately, because of the toxicity of *cisA* expression in *E. coli* and the possibly transient nature of the interaction under the experimental conditions used, we were unable to confirm this interaction by biochemical or biophysical techniques, such as co-purification or bacterial two-hybrid assays. Despite these technical challenges, we believe that the AlphaFold predictions provided a valuable hypothesis about the role of CisA in firing and the function of CIS particles in *S. coelicolor*.

The cisA mutant showed a similarly accelerated sporulation as was previously reported for CIS-negative strains, which supports the conclusion that CisA is required for function of CISSc. But the results do not add any new insights into how CIS/CisA affects the progression of the developmental life cycle and whether this effect has anything to do with the regulated cell death that is caused by CIS. The same applies to the effect on secondary metabolite production, with no further mechanistic insights added, except reporting similar effects of CIS and CisA inactivations.

Thank you for your feedback on this aspect of the manuscript. We would like to note that the main focus of this study was to provide further insight into how CIS contraction and firing are mediated in *Streptomyces*. We used the analysis of accelerated sporulation and secondary metabolite production as a readout to directly assess the functionality of CIS in the presence or absence of CisA and to complement the in situ cryoET data. In summary, our data significantly expand our knowledge of CIS function and firing in *Streptomyces* and suggest a model in which CisA plays an essential role in mediating the interaction of CIS particles with the membrane, which is required for CIS-mediated cell death. We discuss this model in more detail in the revised manuscript (Line 274-283).

We agree that we still don’t fully understand the full nature of the signals that trigger CIS contraction, but we do know that the production of CIS is an integral part of the *Streptomyces* multicellular life cycle as demonstrated by two independent previous studies by us and others (DOI: 10.1038/s41564-023-01341-x and DOI: 10.1038/s41467-023-37087-7).

We further speculate that the assembly and CisA-dependent firing of *Streptomyces* CIS particles could present a molecular mechanism to dismantle part of the vegetative mycelium. This form of “regulated cell death” could provide two key benefits: (1) to prevent the spread of local cellular damage to the rest of mycelium and (2) to provide additional nutrients for the rest of the mycelium to delay the terminal differentiation into spores, which in turn also affects the production of secondary metabolites.

Concluding remarks:The work will be of interest to anyone interested in contractile injection systems, T6SS, or similar machineries, as well for people working on the biology of streptomycetes. There is also a potential impact of the work in the understanding of how such molecular machineries could have been co-opted during evolution to become a mechanism for regulated cell death. However, this latter aspect remains still poorly understood. Even though this paper adds excellent new structural insights and identifies a putative membrane anchor, it remains elusive how the Streptomyces CIS may lead to cell death. It is also unclear what the advantage would be to trigger death of hyphal compartments in response to stress, as well as how such cell death may impact (or accelerate) the developmental progression. Finally, it is inescapable to wonder whether the Streptomyces CIS could have any role in protection against phage infection.

We thank Reviewer #2 for the overall supportive assessment of our work. We will briefly discuss functional CIS's impact on Streptomyces development in the revised manuscript. We previously tested if *Streptomyces* could defend against phages but have not found any experimental evidence to support this idea (unpublished data). The analysis of phage defense mechanisms is an underdeveloped area in *Streptomyces* research, partly due to the currently limited availability of a diverse phage panel.

**Reviewer #3 (Public review):**
Summary:In this work, Casu et al. have reported the characterization of a previously uncharacterized membrane protein CisA encoded in a non-canonical contractile injection system of Streptomyces coelicolor, CISSc, which is a cytosolic CISs significantly distinct from both intracellular membrane-anchored T6SSs and extracellular CISs. The authors have presented the first high-resolution structure of extended CISSc structure. It revealed important structural insights in this conformational state. To further explore how CISSc interacted with cytoplasmic membrane, they further set out to investigate CisA that was previously hypothesized to be the membrane adaptor. However, the structure revealed that it was not associated with CISSc. Using fluorescence microscope and cell fractionation assay, the authors verified that CisA is indeed a membrane-associated protein. They further determined experimentally that CisA had a cytosolic N-terminal domain and a periplasmic C-terminus. The functional analysis of cisA mutant revealed that it is not required for CISSc assembly but is essential for the contraction, as a result, the deletion significantly affects CISSc-mediated cell death upon stress, timely differentiation, as well as secondary metabolite production. Although the work did not resolve the mechanistic detail how CisA interacts with CISSc structure, it provides solid data and a strong foundation for future investigation toward understanding the mechanism of CISSc contraction, and potentially, the relation between the membrane association of CISSc, the sheath contraction and the cell death.Strengths:The paper is well-structured, and the conclusion of the study is supported by solid data and careful data interpretation was presented. The authors provided strong evidence on (1) the high-resolution structure of extended CISSc determined by cryo-EM, and the subsequent comparison with known eCIS structures, which sheds light on both its similarity and different features from other subtypes of eCISs in detail; (2) the topological features of CisA using fluorescence microscopic analysis, cell fractionation and PhoA-LacZα reporter assays, (3) functions of CisA in CISSc-mediated cell death and secondary metabolite production, likely via the regulation of sheath contraction.Weaknesses:(1) The data presented are not sufficient to provide mechanistic details of CisA-mediated CISSc contraction, as authors are not able to experimentally demonstrate the direct interaction between CisA with baseplate complex of CISSc (hypothesized to be via Cis11 by structural modeling), since they could not express cisA in *E. coli* due to its potential toxicity. Therefore, there is a lack of biochemical analysis of direct interaction between CisA and baseplate wedge. In addition, there is no direct evidence showing that CisA is responsible for tethering CISSc to the membrane upon stress, and the spatial and temporal relation between membrane association and contraction remains unclear. Further investigation will be needed to address these questions in future.

We thank Reviewer #3 for the supportive evaluation and constructive feedback of our study in the non-public review. We appreciate the recognition of the technical limitations of experimentally demonstrating a direct interaction between CisA and CIS baseplate complex, and we agree that further investigations in the future will hopefully provide a full mechanistic understanding of the spatiotemporal interaction of CisA and CIS particular and the subsequent CIS firing.

To further improve the manuscript, we will revise the text and clarify figures and figure legends as suggested in the non-public review.

Discussion:Overall, the work provides a valuable contribution to our understanding on the structure of a much less understood subtype of CISs, which is unique compared to both membrane-anchored T6SSs and host-membrane targeting eCISs. Importantly, the work serves as a good foundation to further investigate how the sheath contraction works here. The work contributes to expanding our understanding of the diverse CIS superfamilies.

Thank you.

**Recommendations for the authors:**

**Reviewer #1 (Recommendations for the authors):**
- Magnification of the potential CisA-Cis11 model, with side chains at the interface, should be shown in Supplementary Figures 9/10 to help the reader appreciates the intercation between the two subunits.

Done. A zoomed-in view of the relevant side chains at the CisA-Cis11 interface has been added to Supplementary Figure 9e. For clarity, we decided not to highlight these residues in Supplementary Figure 10 because they are identical to those in Figure 9e.

- A model where CisA is positionned onto the baseplate (by merging the CisA-Cis11 model and the baseplate structure) will also be informative for the reader.

We agree that such a presentation would be helpful to visualize the proposed CisA-Cis11 interaction. However, the Cis11 residues predicted to bind CisA are buried in our cryoEM single-particle structure of the elongated *Streptomyces* CIS. This is not surprising, as the structure is based on a previously established non-contractile CIS mutant variant (PMCID: PMC10066040), which means we were only able to capture one specific configuration of the baseplate complex in the current work. This baseplate configuration is most likely structurally distinct from the baseplate configuration in contracted CIS particles. A similar observation was also reported for the baseplate complex of eCIS particles from *Algoriphagus machipongonesis* (PMCID: PMC8894135).

We speculate that in *Streptomyces*, initial non-specific contacts between CisA and cytoplasmic CIS particles induce a rearrangement of baseplate components, resulting in the exposure of the relevant Cis11 residues, which in turn facilitates a transient interaction between CisA and Cis11. This interaction then leads to additional conformational changes within the baseplate complex, triggering sheath contraction and CIS firing.

We believe that a transient binding step is a crucial part of the activation process, contributing to the dynamic nature of the system.

- Providing information on the oligomeric state of CisA will strenghten the manuscript. Authors may consider having blue-native gel analysis of CisA-3xFLAG extracted from Streptomyces or *E. coli* membranes, or in vivo chemical cross-linking coupled to SDS-PAGE analyses. In case these quite straightforward experiments are not possible, the authors may consider providing AF3 models of various CisA multimers.

Thank you for these suggestions. Unfortunately, we currently don’t have the capability to conduct additional experiments. However, we have performed additional AF3 modelling to explore potential different configurations of CisA. The results of these analyses suggest that CisA can assemble into a pentamer (see also Response to reviewer 1). We speculate that CisA may exist in different oligomeric states and that membrane-localized CisA monomers oligomerize into a larger protein complex in response to a cellular or extracellular (e.g. nisin) signal, which could then directly or indirectly interact with CIS particles in the cytoplasm to facilitate their recruitment to the membrane and CIS firing. Such a stress-dependent conformational change of CisA could also be a safety mechanism to prevent accidental interaction of CisA with CIS particles and CIS firing.

We now show the AF model for the predicted CisA pentamer in Figure 3b/c and discuss the potential implications of the different CisA configurations in the revised manuscript.

**Reviewer #2 (Recommendations for the authors):**
- The quantification of contracted versus extended CIS assemblies in the cytoplasm is only presented for the tomograms from the cisA mutant (graph in Fig. S2d). However, there are no data for the WT and complemented mutant to compare with. It would help to add such data, or at least refer to the previous quantification done for the WT in the previous paper. Further, would it be possible to illustrate the difference by measuring lengths of CIS assemblies and plot length distributions (assuming the extended ones are long and contracted are short)?

Thank you for your suggestions. We have included the results from our previous quantification of CIS assembly states observed in the WT in the revised manuscript (lines 106–110).

In the acquired tomograms of CIS particles observed in intact and dead hyphae, we consistently observed only two CIS conformations: the fully extended state (average length of 233 nm, diameter of 18 nm) and the fully contracted state (average length of 124 nm, diameter of 23 nm). We have added this information to the revised manuscript (lines 112-114).

- The Western blot in Fig. 3d, top panel, contains additional bands that are not mentioned. Are they non-specific bands? Absent in disA mutant? It would help if it was clarified in the legend what they are.

Correct, these additional bands are unspecific bands, which are also visible in the lysate and soluble fraction of wild-type sample (negative control, no FLAG-tagged protein). We have now labelled these bands in the figure and clarified the figure legend.

- Fig. S8a needs improvement. It was not possible to clearly see the stated effect of disA deletion on secondary metabolite production in these photos.

We agree and have removed figure panel S8a from the manuscript. The quantification of total actinorhodin production shown in Figure S8b convincingly shows a significantly reduction of actinorhodin production in the *cisA* deletion mutant compared to the wildtype and the complement mutant.

- It is not an important point, but the paragraph in lines 109-116 appears more like a re-iteration of the Introduction than Results.

We agree. We have removed the highlighted text from the Results section and added some of the information to the introduction.

- Line 206 appears to have a typo. Should it not be WT instead of WT cisA?

Correct. This is a typo which has been fixed. Thank you.

- At the end of the Discussion, it is suggested that a stepwise mechanism of recruiting CIS to the membrane and then triggering firing would prevent unwanted activation and self-inflicted death. Since both steps appear to be dependent in DisA, it would be good to more clearly spell out how such a stepwise mechanism would work and how it could prevent spontaneous and erroneous firing of the system.

Thank you for this suggestion. We have revised the text to clarify the proposed stepwise mechanism. Based on additional structural modeling, we propose that the conserved extra-cytoplasmic domain of CisA may play a role in sensing stress signals. Binding of a ‘stress-associated molecule’ could induce a conformational change in CisA, a hypothesis supported by: (1) Foldseek protein structure searches, which suggest that the conserved C-terminal CisA domain resembles substrate/solute-binding proteins, and (2) AlphaFold3 models predicting that CisA can form a pentamer via its putative substrate-binding domain. This suggests that a transition from CisA monomers to pentamers in response to stress may serve as a key checkpoint, activating CisA and facilitating the recruitment of CIS assemblies to the membrane, either directly or indirectly. Conversely, in the absence of a stress signal, CisA is likely to remain in its monomeric (resting) form, incapable of triggering CIS firing. We have revised the discussion to explain the proposed model in more detail.

We recognize that this model poses many testable hypotheses that we currently cannot test but aim to address in the future.

**Reviewer #3 (Recommendations for the authors):**
There are a few concerns potentially worth addressing to strengthen the study or for future investigation.(1) It would be worth considering moving the first part of the result ('CisA is required for CISSc contraction in situ') after presenting the structure of extended CISSc, and combining it with the last part of the result section ('CisA is essential for the cellular function of CISSc'), as both parts describe the functional characterization of CisA.

We appreciate the reviewer’s suggestion but have chosen to retain the current order of the results. As this manuscript focuses on the role of CisA, we believe that first establishing a functional link between CisA and CIS contraction provides essential context and motivation for the study.

(2) Line 169: it is not clear to me if the fusion of CisA with mCherry is functional (if it complements the native CisA). Moreover, it was not shown if its localization changes under nisin stress or in the strain with non-contractile CISSc.

We have not tested if the CisA-mCherry fusion is fully functional. While we cannot exclude the possibility that the activity of this protein fusion is compromised in vivo, we believe that the described accumulation of CisA-mCherry at the membrane is accurate. This conclusion is further supported by the results obtained from protein fractionation experiments and the membrane topology assay (Figure 3).

We did not examine if the localization of CisA-mCherry changes in CIS mutant strains under nisin-stress, but this is something we will follow up on in the future.

(3) In ref 18, the previous work from the same team presented a functional fluorescent fusion of Cis2 (sheath), thus, it will be interesting to see if (i) Cis2 localization and dynamics is affected by the absence of CisA under normal and stressed conditions; (ii) if Cis2 shows any co-localization with CisA under normal and especially stressed conditions, and potentially, its timing correlation to ghost cell formation by time-lapse imaging of both fusions.

We thank this reviewer for the suggestions, and we plan to address these questions in the future.

(4) Line 261: it was hypothesized by authors that the cytosolic portion of CisA was required for interacting with Cis11. While it was not possible to verify the direct interaction at current state, a S. coelicolor mutant lacking this cytosolic domain may be of help to indirectly test the hypothesis. Moreover, it would be interesting to see if the cytosolic region alone is enough to induce the contraction upon stress (by removing the TM-C region). If so, whether it leads to cell death, or if it is insufficient to cause cell death without membrane association despite the sheath contraction. If not, it would suggest that membrane association occurs before contraction.

These are really great suggestions and if we had the manpower and resources, we would have performed these experiments. We plan to follow up on these questions in the future.

However, additional structural modelling of CisA indicates that CisA may exist in different configurations (see response to Reviewer #1 and #2), a monomeric and/or a pentameric configuration. In these structural models (revised Figure 3), CisA oligomerization is mediated by the annotated periplasmic solute-binding domain. It is conceivable that CisA oligomerization (e.g. in response to a stress signal) presents a critical checkpoint that results in a conformational change within CisA monomers that subsequently drives CisA oligomerization into a configuration primed to interact with CIS particles. We would therefore speculate that the expression of just the cytoplasmic CisA domain may not be sufficient for CIS contraction and cell death.

(5) Line 263: as it was not possible to express full-length cisA in *E. coli*, making it difficult to assess the interaction between CisA and Cis11, it may be worth considering expressing the cytosolic portion of CisA (ΔTM-C) instead of full-length CisA, or alternatively performing a co-immunoprecipitation assay of CisA (i.e., with an affinity tag) from S. coelicolor cultures under stressed conditions. However, I am aware that these may be beyond the scope of this work but can be considered for future investigation in general.

Thank you for your suggestions and your understanding that some of this work is beyond the scope of this work. We have performed CisA-FLAG co-immunoprecipitation experiments from *S. coelicolor* cultures that were treated with nisin for 0/15/45 min. However, mass spectrometry analysis of co-eluted peptides did not show the presence of CIS-associated peptides at the analysed timepoints. While we cannot exclude technical issues with our assays that resulted in an inefficient solubilization of CisA from *Streptomyces* membranes, an alternative hypothesis is that the interaction between CIS particles and CisA is very transient and therefore difficult to capture. We would like to mention, however, that we did detect CisA peptides in crude purifications of CIS particles from nisin-stressed cells (Supplementary Table 2, manuscript: line 301/302), supporting our proposed model that CisA can associate with CIS particles in vivo.

Minor points:(1) I will suggest moving Supplementary Fig 2d with control quantification of WT strain and complementation strain (similar to Fig 3g from ref 18) to the main Fig 1, as the quantitative representation with better comparison without going back and forth to ref 18.

Thank you for your suggestion. Instead of moving Supplementary Fig. 2d to the main figure, we have added additional information in lines 106–110 to discuss the previous quantification of CIS assembly states in the WT, as described in our earlier work. We believe this approach allows readers to easily reference our established quantification without compromising the flow of the main figures.

(2) Line 52/785: as work of Ref 12 has recently been published DOI: 10.1126/sciadv.adp7088, the reference should be updated accordingly.

This reference has been updated. Thank you.

(3) A brief description of key differences between contracted (ref 18) and extended sheath structure will be a good addition for a broader audience.

Thank you for this suggestion. We have added more information on lines 178–180.

(4) Fig 3d: it is not clear how well the samples from different fractions were normalized in amount (volume and cell density), but there was an inconsistency in the amount of CisA-Flag in lysate, vs. soluble and membrane fractions (total protein amount combined from soluble fraction and membrane fraction together seemed to be more than in the lysate, while in theory it should be more or less equal; and the amount of WhiA from WT seemed to be less than from the CisA-Flag strain). In the method section, it was mentioned that 'The final pellet was dissolved in 1/10 of the initial volume with wash buffer (no urea). Equi-volume amounts of fractions were mixed with 2x SDS sample buffer and analyzed by immunoblotting.' But it is still not clear whether equivalent amounts (normalized to the same OD for example) were used and if we could directly compare. A brief clarification in the legend of how samples were prepared is needed.

The samples were normalized by first using the same volume of starting material (similar culture density and incubation period for each strain) and by loading equal volumes of each fraction for analysis. After fractionation, equi-volume amounts of the soluble and membrane protein fractions were mixed with 2× SDS sample buffer and subjected to immunoblotting, ensuring a consistent basis for comparison between samples. We have revised the figure legend and Material and Method sections to make this clear.

We agree that the amount of CisA-3xFLAG appears slightly lower in the “Lysate” fraction compared to the “Membrane” fraction in Figure 3d (now Fig. 3f). However, this does not affect the overall conclusion of this experiment, showing that CisA-3xFLAG is clearly enriched in the membrane fraction.

For reference, please find below the uncropped version of this Western blot image. Based on the signal of the unspecific bands, we would like to argue that equal amounts of samples obtained from the WT control strain (no FLAG epitope present) and a strain producing CisA-3xFLAG were loaded for each of the fractions. When we revisited this data, we noted that the protein size marker was wrong. This has been fixed.

(5) Fig. 4f: statistical analysis is missing.

The missing statistical analysis has been added to this figure and figure legend.